# FreeKV: Boosting KV Cache Retrieval for Efficient LLM Inference

**Guangda Liu**[1]    **Chengwei Li**[1]    **Zhenyu Ning**[1]    **Jing Lin**[2]    **Yiwu Yao**[2]    **Danning Ke**[2]
**Minyi Guo**[1]    **Jieru Zhao**[1]*
[1] School of Computer Science, Shanghai Jiao Tong University    [2] Huawei Technologies Co., Ltd

## Abstract

Large language models (LLMs) are widely deployed with rapidly expanding context windows to support increasingly demanding applications. However, long contexts pose significant deployment challenges, primarily due to the KV cache whose size grows proportionally with context length. While KV cache compression methods have been proposed to address this issue, KV dropping methods incur considerable accuracy loss, and KV retrieval methods suffer from significant efficiency bottlenecks. We propose **FreeKV**, a training-free algorithm-system co-optimization framework to enhance KV retrieval efficiency while preserving accuracy. On the algorithm side, FreeKV introduces speculative retrieval to shift the KV selection and recall processes out of the critical path, combined with fine-grained correction to ensure accuracy. On the system side, FreeKV employs hybrid KV layouts across CPU and GPU memory to eliminate fragmented data transfers, and leverages double-buffered streamed recall to further improve efficiency, enabling effective overlap with computation, full latency hiding, and practical speedups from speculative recall. Experiments demonstrate that FreeKV achieves near-lossless accuracy across various scenarios and models, delivering up to a $13\times$ speedup compared to SOTA KV retrieval methods. Code is available at https://github.com/sjtu-zhao-lab/FreeKV.

## 1 Introduction

Large language models (LLMs) have gained remarkable prominence for their ability to excel across diverse tasks and have been widely deployed in a variety of applications, such as document analysis, chatbots and coding assistants (Chew et al., 2023; Jiang et al., 2024b). To process increasingly complex tasks such as long-document QA, multi-turn dialogue, and repository-level code understanding, the context window sizes of LLMs are rapidly expanding to accommodate longer inputs. Mainstream LLMs now support context windows of 128K tokens (Qwen et al., 2025; DeepSeek-AI et al., 2025), with frontier models reaching up to 1 million tokens (DeepMind, 2025; xAI, 2025).

While larger context windows unlock new capabilities for applications, handling long context presents significant challenges for efficient deployment. These challenges arise from the KV cache in LLMs, which stores the key-value states of previous tokens to avoid recomputation during inference, causing its size to grow proportionally with the context length. On the one hand, the size of KV cache can exceed the capacity of GPU memory. For instance, the KV cache for a single request can reach 40GB for Llama-3-70B with a context length of 128K (Meta, 2024). On the other hand, since the LLM decoding is memory-bound, accessing a large KV cache significantly degrades the decoding speed (Fu, 2024).

To mitigate these issues, based on the sparsity of attention computation, previous works proposed compressing the KV cache, i.e., utilizing only a portion of the KV cache for inference. These compression methods can be broadly classified into two categories: **KV dropping** and **KV retrieval** (Li et al., 2025). KV dropping methods only retain KV cache for important tokens and permanently evict unimportant ones. The identification of important tokens can be performed either statically (Xiao et al., 2024a) or dynamically (Li et al., 2024). In contrast, KV retrieval methods maintain the entire KV cache but dynamically select a subset for inference (Chen et al., 2024b; Liu et al., 2024).

---

*Correspondence to Jieru Zhao (zhao-jieru@sjtu.edu.cn)

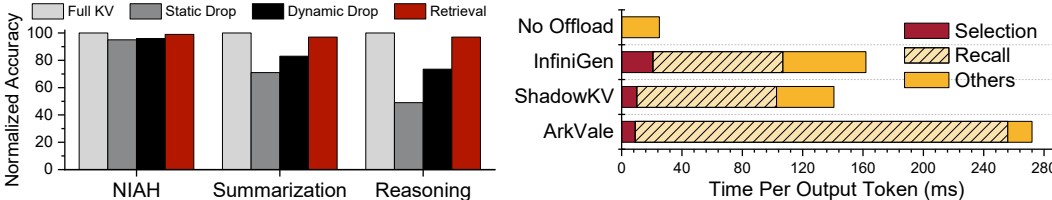

Figure 1: Left: Accuracy comparison of KV dropping and retrieval methods across different tasks. Right: Latency breakdown of KV retrieval methods with offloading.

While both KV dropping and retrieval methods can maintain acceptable model accuracy under specific scenarios and tasks, recent studies reveal significant accuracy degradation with KV dropping methods, particularly on tasks like summarization and reasoning (Gao et al., 2025a; Liu et al., 2025). This degradation stems from the dynamic nature of token importance, where tokens previously deemed unimportant and permanently dropped may become crucial in later steps (Chen et al., 2024a; Liu et al., 2024). For complex tasks involving long generation, the omission of a large number of such important tokens results in severe accuracy decline. This issue is further exacerbated with the advent of reasoning models, where extended thinking processes lead to generation lengths reaching 32K tokens or more (OpenAI, 2024; DeepSeek-AI et al., 2025; Qwen, 2025b), highlighting the limitations of KV dropping methods. In Fig. 1, we compare the accuracy of static KV dropping (RazorAttention (Tang et al., 2024a)), dynamic KV dropping (RaaS (Hu et al., 2025)) and KV retrieval (Quest (Tang et al., 2024b)) under similar KV cache budgets, across tasks of Needle In A Haystack (NIAH), summarization and reasoning (Kamradt, 2023; Bai et al., 2023; Huggingface, 2025). Both static and dynamic drop methods exhibit significant accuracy degradation on summarization and reasoning tasks. In contrast, KV retrieval methods maintain robust accuracy across all tasks. Therefore, **KV retrieval methods are better suited for more general and practical scenarios.**

Despite their superior accuracy performance, **KV retrieval methods face significant efficiency challenges**. First, since the complete KV cache must be retained, retrieval methods often offload the KV cache to CPU memory to circumvent GPU memory limitations. For methods without offloading like Quest, out-of-memory errors are inevitable for long contexts and large batch sizes. However, for offloading methods, due to the low bandwidth of the CPU-GPU connection, *recalling* the selected KV tuples from CPU to GPU memory incurs long latency. Second, KV retrieval methods select KV tuples from the entire context, leading to considerable *selection* overhead, even though most retrieval methods adopt page-wise selection to alleviate this issue. In Fig. 1, we present the latency breakdown of SOTA offloading methods, using Llama-3.1-8B-Instruct with a batch size of 1 and a context length of 32K. For ArkVale (Chen et al., 2024a), recall and selection contribute approximately 94% of the overall latency. Similarly, while ShadowKV (Sun et al., 2025) introduces key cache reconstruction to recall only the value cache, recall and selection still comprise about 73% of the total latency. InfiniGen (Lee et al., 2024) overlaps recall with computation by re-projecting the query and key vectors to prefetch the KV cache for the next layer. However, as shown in Fig. 1, InfiniGen's recall latency cannot be fully hidden due to its inefficient token-wise recall, with the remaining unoverlapped portion still accounting for nearly 53% of the overall latency. Moreover, InfiniGen incurs substantial overhead from the re-projection and selection. All these retrieval methods lead to significantly higher latency compared to inference with the full KV cache without offloading.

To overcome these challenges, we introduce FreeKV, a training-free algorithm-system co-optimization framework that significantly boosts the efficiency of KV retrieval, while maintaining near-lossless model accuracy across diverse scenarios. **On the algorithm side**, leveraging the high similarity of query vectors between adjacent decoding steps, FreeKV introduces *speculative retrieval*, which shifts the selection and recall processes out of the critical path via step-wise KV reuse, thus avoiding inference blocking. As shown in Fig. 2a, this approach allows selection and recall to overlap with other operations, effectively hiding their overhead. To counter potential accuracy losses from pure KV reuse, FreeKV incorporates *fine-grained correction* to preserve model accuracy with minimal impact on efficiency. **On the system side**, FreeKV employs *hybrid KV layouts* across CPU and GPU memory to eliminate inefficient fragmented data transfers and avoid layout conversion overhead during inference. In addition, FreeKV implements a double-buffering mechanism to facilitate *streamed recall*, further improving recall efficiency by overlapping CPU-GPU and GPU-GPU

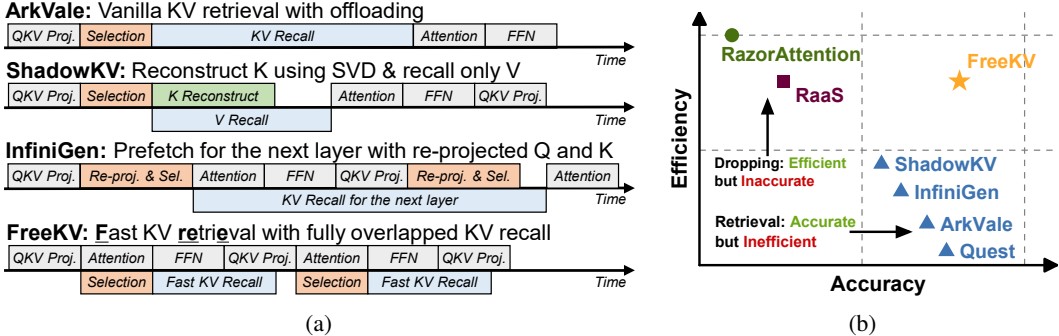

Figure 2: (a) Comparison of timelines for KV retrieval methods, FreeKV shifts the selection and recall out of the critical path. (b) Accuracy-efficiency trade-off of KV compression methods.

data transfers. These system-side optimizations dramatically reduce recall latency, enabling effective overlap with computation, full latency hiding, and practical speedups from speculative recall. As shown in Fig. 2b, FreeKV strikes a balance between accuracy and efficiency, establishing a new Pareto frontier. Extensive experiments show that FreeKV maintains near-lossless accuracy across diverse scenarios and models, delivering up to $13\times$ speedup over SOTA KV retrieval methods.

## 2 BACKGROUND AND RELATED WORK

### 2.1 PROBLEM FORMULATION

The decoding process of an attention head in LLMs can be expressed as $\mathbf{o} = \text{softmax}(\mathbf{q}\mathbf{K}^T/\sqrt{d})\mathbf{V}$, where $\mathbf{q}, \mathbf{o} \in \mathbb{R}^{1\times d}$ are the query vector and attention output of the current token, respectively, and $\mathbf{K}, \mathbf{V} \in \mathbb{R}^{L\times d}$ represent the K and V states of the $L$ preceding tokens. For modern LLMs with Grouped Query Attention (GQA) (Ainslie et al., 2023), the number of attention heads and KV heads are denoted as $n_{qo}$ and $n_{kv}$, respectively. The output of attention head $h$ is given by $\mathbf{o}_h = \text{softmax}(\mathbf{q}_h\mathbf{K}_\hbar^T/\sqrt{d})\mathbf{V}_\hbar$, where $\hbar = \frac{h}{n_{qo}/n_{kv}} = \frac{h}{G}$ is the corresponding KV head. And $G = \frac{n_{qo}}{n_{kv}}$ is the group size, representing the number of attention heads within a group that share the same KV head.

KV retrieval methods select a subset of KV tuples for attention computation based on attention weights derived from $\mathbf{q}$ and $\mathbf{K}$, defined as $\mathcal{I}^h = Sel(\mathbf{q}^h, \mathbf{K}^\hbar)$, where $\mathcal{I}^h$ represents the indices of selected KV tuples, and $|\mathcal{I}^h| = \mathcal{B}$ specifies a preset KV cache budget. In practice, retrieval methods consistently retain KV tuples for $\mathcal{S}$ sink tokens at the beginning and $\mathcal{W}$ tokens within the local window, leaving $\mathcal{B} - \mathcal{S} - \mathcal{W}$ tuples available for selection.

For GQA models, the space required for retrieved KV tuples is $O(\mathcal{B} \times n_{kv})$ if the selection is *group-consistent*, meaning the indices of KV tuples selected by all attention heads within the same group are identical, i.e., $\mathcal{I}^{(m-1)\times G+1} = \mathcal{I}^{(m-1)\times G+2} = \cdots = \mathcal{I}^{m\times G}$ for $m = 1, 2, \ldots, n_{kv}$. If the selection is not group-consistent, the required space increases to $O(\mathcal{B} \times n_{qo})$, resulting in $G$ times higher costs in both space and memory accesses.

### 2.2 RELATED WORK

**KV dropping** KV dropping methods can be further categorized into static and dynamic dropping. Static dropping methods evict KV states using fixed patterns determined before inference. For instance, StreamingLLM (Xiao et al., 2024b) retains KV tuples only for the initial *sink* tokens and those within a local window. Based on this, RazorAttention (Tang et al., 2024a) and DuoAttention (Xiao et al., 2024a) retain full KV cache for designated *retrieval heads*, while limiting other heads to sink tokens and a local window. Static dropping methods are computationally efficient, incurring minimal overhead during inference, with GPU memory usage proportional to a preset sparsity $\mathbf{s}$ and the context length $L$. However, their fixed nature overlooks dynamic patterns during inference, leading to significant accuracy losses. Dynamic dropping methods, on the other hand,

Table 1: Comparison of KV cache compression methods.

| | RazorAttn | RaaS | Quest | ArkVale | ShadowKV | InfiniGen | FreeKV |
|---|---|---|---|---|---|---|---|
| Category | Static Drop | Dynamic Drop | Retrieval | Retrieval | Retrieval | Retrieval | Retrieval |
| Long Generation | ✔ | ✔ | ✔ | ✔ | ✗ | ✗ | ✔ |
| GPU Mem. Usage | $O(\mathbf{s}L)$ | $O(\mathcal{B})$ | $O(L)$ | $O(\mathcal{B})$ | $O(\frac{r}{d_{kv}}L+\mathcal{B})$ | $O(\frac{r}{d_{kv}}L+\mathcal{B})$ | $O(\mathcal{B})$ |
| Group-consistent | ✔ | ✗ | ✗ | ✔ | ✔ | ✗ | ✔ |
| Efficiency | High | High | No Offload | Slow Recall No Overlap | SVD Cost Pool Overlap | Re-proj. Cost Pool Overlap | Fast Recall Full Overlap |

evict KV tuples based on attention scores calculated online during inference (Zhang et al., 2023; Li et al., 2024). While most dropping methods do not support the long-generation scenarios, RaaS (Hu et al., 2025) addresses these scenarios by evicting tokens that have not received significant attention scores for a sustained period. Dynamic dropping methods retain and score only a fixed budget of the KV cache, achieving good efficiency despite the additional scoring overhead.

**KV retrieval** Both static and dynamic dropping methods incur permanent information losses, resulting in notable accuracy degradation, particularly in long-generation scenarios. In contrast, KV retrieval methods retain the complete KV cache but select a subset for computation. While preserving accuracy, retrieval methods introduce significant efficiency challenges. First, **applying selection across the entire context by scoring over every token leads to unacceptable overhead**. To mitigate this, most KV retrieval methods adopt **page-wise selection**, summarizing the keys within a page and scoring only these *page summaries*. For example, Quest uses min-max pooled keys, ArkVale employs bounding volumes of keys within a page, and ShadowKV simply relies on mean-pooled keys. Moreover, **the handling of the complete KV cache poses significant challenges**. Quest stores the entire KV cache in GPU memory with limited capacity, restricting support for long context lengths and large batch sizes. Furthermore, its inconsistent selection within head groups incurs $G$ times memory access overhead. ArkVale offloads the KV cache to CPU memory and recalls the selected KV pages during inference. While ensuring group-consistent using mean pooling over attention weights and maintaining a cache for selected pages on GPU, the recall process of ArkVale remains costly, severely impacting efficiency. ShadowKV takes a different approach by leveraging the low-rank property of the pre-rope key cache. It retains only the low-rank key cache obtained through singular value decomposition (SVD). During inference, for selected pages, it reconstructs the key cache from the low-rank representations, while only recalling the value cache. This reduces memory transfer costs but requires additional GPU memory to store the low-rank key, consuming $\frac{r}{d_{kv}}$ (15%-30%) of the original key cache size, where $r = 160$ is the rank used by ShadowKV and $d_{kv}$ is the dimension of key cache. Moreover, ShadowKV does not support long-generation since the SVD is performed only once during prefill, leaving the low-rank key unupdated during decoding. Leveraging the high similarity of hidden states across adjacent layers, InfiniGen re-projects the current layer's hidden states using the skewed query weights of the next layer, to pre-select and prefetch the KV cache for the next layer. While this strategy enables partial overlap between recall and computation, InfiniGen's recall latency cannot be fully hidden due to its inefficient token-wise recall. In addition, InfiniGen incurs substantial computation and memory overhead. An extra $\frac{r}{d_{kv}}$ (30%) skewed key cache must be computed and maintained for all tokens, and a skewed query vector must be computed at every decoding step.

The features of KV dropping and retrieval methods are summarized in Table 1. As illustrated, FreeKV ensures accuracy preservation through KV retrieval, while attaining high efficiency with fixed $O(\mathcal{B})$ GPU memory usage, group-consistent selection, and fully overlapped, efficient recall.

## 3 ALGORITHM DESIGN

### 3.1 OBSERVATION

We sample the query vectors of generated tokens during inference of Llama-3.1-8B-Instruct on long-generation tasks and DeepSeek-R1-Qwen-14B on long-reasoning tasks. The cosine similarity between the query vectors of adjacent generated tokens is calculated as $\mathcal{C}_i = \frac{\langle \mathbf{q}_i, \mathbf{q}_{i-1} \rangle}{|\mathbf{q}_i| \cdot |\mathbf{q}_{i-1}|}$, where $\mathbf{q}_{i-1}, \mathbf{q}_i \in \mathbb{R}^{1 \times d}$ are the query vectors of tokens generated at step $i-1$ and $i$. We present the mean

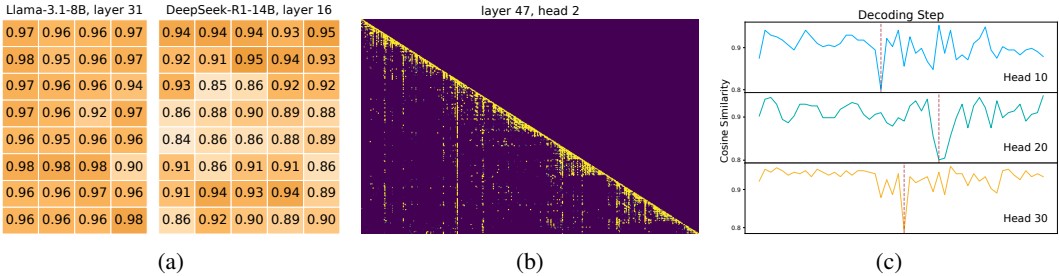

Figure 3: (a) Cosine similarities between query vectors of adjacent generated tokens, averaged over generation; each cell corresponds to an attention head. (b) Attention map of DeepSeek-R1-Qwen-14B on reasoning tasks. (c) Variations in $\mathcal{C}_i$ during generation of DeepSeek-R1-Qwen-14B.

similarity during generating $g = 13000$ tokens, calculated as $\sum_{i=1}^{g} \frac{\mathcal{C}_i}{g}$, in Fig. 3a. As shown, across various models and tasks, the mean similarity of all attention heads consistently exceeds 0.84, with most heads achieving a similarity greater than 0.9. This high similarity is observed across different layers, models, and tasks, likely due to position embeddings (Su et al., 2023) and the semantic continuity of adjacent tokens (Yuan et al., 2025). This observation aligns with the vertical line patterns in attention maps, as shown in Fig. 3b and supported by prior studies (Jiang et al., 2024a; Hu et al., 2025; Wu et al., 2025), which show that adjacent decoding steps exhibit high attention scores on similar tokens. This insight motivates our **speculative recall** mechanism (Sec. 3.2).

To delve deeper, we analyze the changes in similarity during generation for DeepSeek-R1-Qwen-14B on reasoning tasks. As shown in Fig. 3c, while the mean similarity remains high, certain decoding steps exhibit outliers with significantly lower similarity. Moreover, these outlier steps vary across attention heads, indicating head-specific variations in query similarity during decoding. These variations underpins the **fine-grained correction** mechanism (Sec. 3.3) employed by FreeKV.

## 3.2 SPECULATIVE RETRIEVAL

**Speculative retrieval**  Based on the high similarity observed in query vectors and selected tokens between adjacent decoding steps, i.e., $Sel(\mathbf{q}_i, \mathbf{K}) \sim Sel(\mathbf{q}_{i-1}, \mathbf{K})$, we propose a speculative retrieval mechanism that shifts the selection and recall out of the critical path of inference. Specifically, the attention computation of step $i$ bypasses the selection and recall, instead directly being launched by reusing the KV tuples recalled during step $i - 1$, as shown in Fig. 4a. This design enables the selection and recall operations to overlap with attention and FFN computations of the current layer, as well as the QKV projections of the next layer. The recalled KV tuples during step $i$ will then be reused in step $i + 1$, continuing the process iteratively.

Compared to InfiniGen, speculative retrieval hides the latency of both selection and recall without incurring additional overhead, eliminating the need for re-projection. We further compare speculative retrieval with direct recall using the last layer's query vector in Appendix B.1, which demonstrates that leveraging the last step's query vector in speculative retrieval yields superior accuracy.

**Group-consistent selection**  FreeKV adopts page-wise selection, utilizing the min-max pooled keys within each page as the page summary, similar to Quest (Tang et al., 2024b). Let $n_{\text{page}}$ denote the number of KV pages. To ensure group-consistent selection, after computing the attention weights $\mathcal{P}^h \in \mathbb{R}^{n_{\text{page}}}$ for query vector of attention head $h$ and the corresponding page summaries, FreeKV applies mean pooling across the group over $\text{softmax}(\mathcal{P}^h)$. For KV head $m$, the corresponding attention heads select consistent pages based on scores calculated as $\sum_{j=1}^{G} \text{softmax}(\mathcal{P}^{(m-1) \times G+j})/G$. Other alternatives for achieving group-consistent selection are evaluated in Appendix B.2, and the results show that mean pooling over the softmax of page attention weights yields the best performance.

## 3.3 FINE-GRAINED CORRECTION

While purely reusing KV pages recalled from the previous step maximizes efficiency, it can result in significant accuracy degradation. To mitigate this, FreeKV introduces a correction mechanism

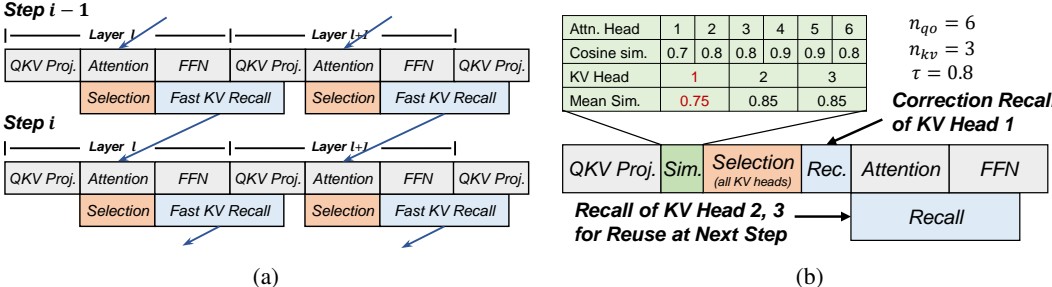

Figure 4: (a) Inference timeline with speculative retrieval, where the blue arrows represent the reuse of KV pages recalled in the previous step. (b) Timeline for fine-grained correction with query-based identification and head-wise correction recall.

that selectively recall KV pages for the current step. By employing query-based identification and head-wise recall, FreeKV minimizes the associated efficiency overhead.

**Query-based identification**  A straightforward correction involves directly comparing the indices of selected KV tuples between step $i$ and $i-1$, i.e., $Sel(\mathbf{q}_i, \mathbf{K})$ and $Sel(\mathbf{q}_{i-1}, \mathbf{K})$. However, this approach incurs substantial overhead due to index comparisons and hinders the overlap of selection with other operations. To address these limitations, FreeKV employs a correction mechanism based on the cosine similarity of query vectors, $\mathcal{C}_i$. Correction is triggered only if $\mathcal{C}_i < \tau$, indicating a significant deviation of $Sel(\mathbf{q}_i, \mathbf{K})$ from $Sel(\mathbf{q}_{i-1}, \mathbf{K})$, where $\tau$ is a predefined threshold. To ensure group consistency, FreeKV performs mean pooling over $\mathcal{C}_i$ across the group, and compares the pooled value with $\tau$ to determine whether correction is required for a KV head. As illustrated in Fig. 4b, KV head 1, with a mean similarity of 0.75 (below $\tau = 0.8$), is flagged for correction. In Appendix B.3, we compare mean pooling with max pooling for achieving group-consistent correction, and show that mean pooling is the more effective choice. We further evaluate different threshold settings and identify the optimal values for different scenarios in Appendix B.3.

**Head-wise correction**  As shown in Fig 4b, once the KV heads requiring correction are identified, FreeKV initiates selection and recall for these KV heads before the attention computation at the current decoding step. For KV heads that do not require correction, recall is deferred and overlapped with other operations, retrieving selected KV tuples for the reuse at the next decoding step. To avoid the overhead and reduce GPU utilization caused by separately launching selection for corrected and non-corrected heads, FreeKV executes selection for all KV heads whenever correction is required. Then for non-corrected KV heads, the recall proceeds directly without repeating the selection.

## 4  SYSTEM DESIGN AND IMPLEMENTATION

Effective recall overlapping to minimize overhead demands high recall efficiency. FreeKV achieves this through a dedicated system design and implementation, featuring caching, hybrid layouts and streamed recall, as detailed in the following sections.

### 4.1  OVERVIEW

The system overview of FreeKV is illustrated in Fig. 5. In the data plane, FreeKV retains the query vectors from the previous step, page summaries and cache for selected KV pages in GPU memory. In CPU memory, FreeKV maintains a complete KV cache pool for offloading KV pages. In the control plane, a controller on CPU manages the scheduling and synchronization of operations such as correction, attention, selection and recall launched on different CPU threads and GPU streams, following the timeline described in Sec. 3.

### 4.2  HYBRID LAYOUTS AND STREAMED RECALL

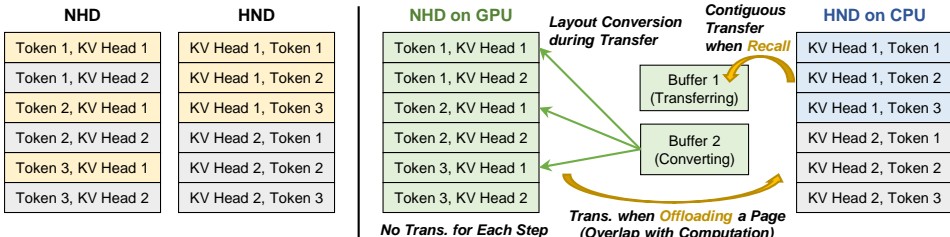

Figure 6: Left: KV cache pages under `NHD` and `HND` layouts with $p = 3$ and $n_{kv} = 2$; the highlights represent elements for a given KV head. Right: Hybrid KV cache layouts across CPU and GPU memory, along with streamed recall enabled by double-buffering.

The KV cache layout defines the memory organization of the underlying key-value tensors. Two commonly used KV cache layouts are `NHD` and `HND` (flashinfer ai, 2025). The `NHD` layout organizes the KV cache in the shape of $(L, n_{kv}, d)$, while the `HND` layout uses the shape of $(n_{kv}, L, d)$. In practice, when managing the KV cache in pages, the shapes of `NHD` and `HND` layouts are $(n_{\text{page}}, p, n_{kv}, d)$ and $(n_{\text{page}}, n_{kv}, p, d)$, respectively, where $p$ is the page size. Since the key and value derived from projections over hidden states are

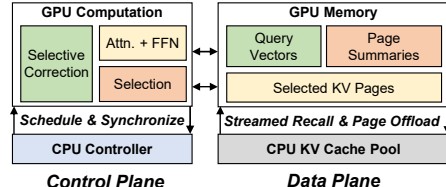

Figure 5: System overview of FreeKV.

$K, V \in \mathbb{R}^{L \times (n_{kv} \times d)}$, `NHD` is the natural layout while the `HND` layout requires additional transpose operations. To eliminate this overhead, mainstream efficient inference frameworks adopts the `NHD` layout (Dao, 2024).

However, since the indices of selected KV pages differ across KV heads and recall is performed individually for each KV head, using the `NHD` layout results in inefficient fragmented data transfers. As shown on the left side of Fig. 6, under the `NHD` layout, for a given KV head, the memory of $p = 3$ key/value vectors within a page is non-contiguous. When recalling a key/value page, the maximum transfer unit contains only $d$ elements, equivalent to just 256 bytes for $d = 128$ and Float16 precision. This extensive fragmented data transfers significantly degrade recall efficiency. In contrast, the `HND` layout ensures that $p$ key/value vectors within a page are contiguous for each KV head, allowing a transfer unit of $p \times d$ elements, or 8KB when $p = 32$.

**Hybrid layouts**  To avoid fragmented data transfer while minimizing transpose overhead, FreeKV adopts hybrid layouts on CPU and GPU memory. As shown in Fig. 6, FreeKV employs the `NHD` layout on GPU to eliminate the need for per-step transposes during decoding, and the `HND` layout on CPU to ensure contiguous and efficient CPU-GPU data transfers during recall. With the hybrid layouts, the `NHD`-`HND` transpose is only required when offloading a KV page, effectively amortizing the overhead. In addition, FreeKV utilizes an `HND` layout on CPU with a shape of $(n_{\text{page}}, n_{kv}, 2, p, d)$, enabling the transfer of $2 \times p \times d$ contiguous elements for both key and value vectors during recall.

**Streamed recall**  While offloading and the associated transposes can overlap with computation, the conversion from `HND` layout to `NHD` layout during recall can block data transfers and subsequent attention computation. To avoid such blocking from sequential data transfer and layout conversion, FreeKV employs a *double-buffering* mechanism to achieve streamed recall. As shown in Fig. 6, after a selected KV page is transferred to buffer 2, its layout conversion begins immediately, while the transfer of the next page is concurrently initiated into buffer 1. Both buffers and the conversion process reside in GPU memory, leveraging its high bandwidth to enhance efficiency.

Table 2: Accuracy results of LongBench v2 and LongGenBench.

| Benchmark | Metric | Full | Razor | RaaS | Quest | ArkVale | ShadowKV | InfiniGen | FreeKV |
|---|---|---|---|---|---|---|---|---|---|
| | | | | *Llama-3.1-8B-Instruct* | | | | | |
| **LongBench v2** | **Overall** | 29.22 | 27.44 | 28.23 | 28.43 | 28.63 ② | 25.45 | 28.56 | 29.22 ① |
| | **Short** | 34.44 | 33.89 | 33.89 | 33.33 | 33.89 ② | 32.78 | 32.28 | 35.00 ① |
| | **Medium** | 27.91 | 25.12 | 26.51 | 27.44 ② | 26.98 | 22.79 | 26.05 | 27.44 ① |
| | **Long** | 23.15 | 21.30 | 22.02 | 22.22 | 23.15 | 18.52 | 24.07 ① | 23.15 ② |
| **LongGenBench** | **CR** | 80.03 | 35.90 | 76.63 | 78.03 | 39.36 | 79.28 ② | 76.68 | 78.03 ② |
| | **CR×Acc** | 26.82 | 12.20 | 26.00 | 27.71 ② | 10.36 | 30.66 ① | 26.21 | 27.62 |
| | | | | *Qwen-2.5-7B-Instruct* | | | | | |
| **LongBench v2** | **Overall** | 27.44 | 25.25 | 26.24 | 27.63 ① | 26.84 | 25.84 | 26.44 | 26.84 ② |
| | **Short** | 36.11 | 32.78 | 35.56 | 36.67 ① | 36.11 ② | 32.22 | 32.22 | 34.44 |
| | **Medium** | 23.72 | 21.86 | 21.86 | 22.79 ② | 22.33 | 20.00 | 23.26 ① | 22.33 |
| | **Long** | 20.37 | 19.44 | 19.44 | 22.22 | 20.37 | 26.85 ① | 23.15 | 23.15 ② |
| **LongGenBench** | **CR** | 79.56 | 42.13 | 77.65 ① | 62.89 | 75.91 | 35.49 | 72.96 | 76.93 ② |
| | **CR×Acc** | 31.09 | 21.48 | 34.40 ① | 25.96 | 31.79 | 11.43 | 27.67 | 32.81 ② |
| | | | | *Qwen-2.5-14B-Instruct* | | | | | |
| **LongBench v2** | **Overall** | 33.40 | 34.19 | 32.60 | 33.80 | 34.19 | 34.79 ① | 32.31 | 34.19 ② |
| | **Short** | 41.11 | 43.33 ① | 40.56 | 40.00 | 41.11 | 40.56 | 40.56 | 41.11 ② |
| | **Medium** | 31.16 | 30.70 | 32.09 | 33.49 | 33.49 | 34.88 ① | 29.89 | 33.49 ② |
| | **Long** | 25.00 | 25.93 ① | 20.37 | 24.07 | 24.07 | 25.00 ② | 23.95 | 24.07 |
| **LongGenBench** | **CR** | 65.84 | 26.48 | 62.29 | 45.49 | 45.31 | 21.25 | 63.02 ② | 65.46 ① |
| | **CR×Acc** | 29.35 | 13.89 | 29.79 ① | 19.76 | 19.65 | 8.25 | 26.08 | 29.39 ② |

## 5 EVALUATION

### 5.1 EXPERIMENTAL SETUP

**Datasets and models** We evaluate FreeKV across various models and tasks. For accuracy evaluation, we select LongBench v2 (Bai et al., 2024) and LongGenBench (Wu et al., 2024) to cover general long-input and long-generation scenarios. In addition, we assess FreeKV on long reasoning tasks, including MATH500 (Hendrycks et al., 2021), AIME24 (Huggingface, 2025) and GPQA (Rein et al., 2023). For LongBench v2 and LongGenBench, we use general models including Llama-3.1-8B-Instruct, Qwen-2.5-7B-Instruct and Qwen-2.5-14B-Instruct. For reasoning tasks, we use DeepSeek-R1-Llama-8B, DeepSeek-R1-Qwen-7B and DeepSeek-R1-Qwen-14B (DeepSeek-AI et al., 2025). Detailed metrics of each dataset are provided in the corresponding sections.

**Baselines** We compare FreeKV against SOTA methods, including KV dropping methods such as RazorAttention and RaaS, as well as KV retrieval methods, including Quest, ArkVale, ShadowKV, and InfiniGen. The sparsity of RazorAttention is set to 0.15, while the budget $\mathcal{B}$ for all other methods is consistently set to 2048. Detailed settings of all methods are provided in Appendix A.

### 5.2 ACCURACY EVALUATION

**LongBench v2** Improved from LongBench (Bai et al., 2023), LongBench v2 covers more realistic scenarios. It spans various difficulty levels and context lengths, ranging from 8K to 2M tokens. All problems of LongBench v2 are presented in a multi-choices question format, with accuracy used as the unified metric. We report accuracy under the context length categories of *short*, *medium* and *long*, as well as the overall accuracy. For all methods, we truncated the inputs to 64K tokens

As shown in Table 2, for all models, the overall accuracy of FreeKV deviates by at most 0.6 compared to the model with full KV cache, while FreeKV achieves the best or second-best performance across most metrics. KV dropping methods, although exhibiting moderate accuracy losses on this long-input benchmark, consistently underperform compared to KV retrieval methods.

**LongGenBench** Unlike traditional long-context benchmarks that focus on long inputs, LongGenBench is designed to evaluate the model's ability to handle long generations, assessing the capability to generate coherent and high-quality long-form content. Each task of LongGenBench contains subtasks that prompt the model to generate specific content at designated points, within specific ranges or in a periodic manner. We report the completion rate (**CR**) of subtasks and the overall

Table 3: Accuracy results of long reasoning tasks.

| Benchmark | Metric | Full | Razor | RaaS | Quest | ArkVale | ShadowKV | InfiniGen | FreeKV |
|---|---|---|---|---|---|---|---|---|---|
| *DeepSeek-R1-Llama-8B* | | | | | | | | | |
| MATH500 | pass@$k$ | 78.00 | 72.00 | 74.00 | 72.00 | 72.00 | 76.00 | 74.00 | 78.00 |
| | avg@$k$ | 67.25 | 60.50 | 62.50 | 62.00 | 62.50 | 60.25 | 66.50 | 66.75 |
| AIME24 | pass@$k$ | 80.00 | 46.67 | 66.67 | 73.33 | 80.00 | 63.33 | 70.00 | 76.67 |
| | avg@$k$ | 47.08 | 30.00 | 36.25 | 44.17 | 46.67 | 36.50 | 45.83 | 47.50 |
| GPQA | pass@$k$ | 82.00 | 60.00 | 64.00 | 76.00 | 72.00 | 78.00 | 46.00 | 86.00 |
| | avg@$k$ | 39.75 | 34.25 | 33.50 | 37.25 | 39.75 | 36.25 | 27.50 | 41.25 |
| *DeepSeek-R1-Qwen-7B* | | | | | | | | | |
| MATH500 | pass@$k$ | 78.00 | 72.00 | 74.00 | 76.00 | 76.00 | 74.00 | 74.00 | 78.00 |
| | avg@$k$ | 71.75 | 66.75 | 67.00 | 68.00 | 68.25 | 64.75 | 70.00 | 70.00 |
| AIME24 | pass@$k$ | 83.33 | 65.33 | 73.33 | 76.67 | 73.33 | 73.33 | 63.33 | 83.33 |
| | avg@$k$ | 56.66 | 35.42 | 42.92 | 47.50 | 47.92 | 43.75 | 43.34 | 52.92 |
| GPQA | pass@$k$ | 72.00 | 60.00 | 58.00 | 72.00 | 72.00 | 70.00 | 58.00 | 74.00 |
| | avg@$k$ | 35.75 | 32.50 | 33.25 | 38.75 | 34.25 | 33.50 | 35.50 | 39.50 |
| *DeepSeek-R1-Qwen-14B* | | | | | | | | | |
| MATH500 | pass@$k$ | 74.00 | 70.00 | 68.00 | 76.00 | 72.00 | 76.00 | 72.00 | 78.00 |
| | avg@$k$ | 70.25 | 59.75 | 64.75 | 67.25 | 66.25 | 65.00 | 67.00 | 67.50 |
| AIME24 | pass@$k$ | 86.67 | 46.67 | 73.33 | 83.33 | 76.67 | 83.33 | 76.67 | 83.33 |
| | avg@$k$ | 66.25 | 32.50 | 48.75 | 58.33 | 61.25 | 57.25 | 60.00 | 64.17 |
| GPQA | pass@$k$ | 82.00 | 68.00 | 80.00 | 80.00 | 86.00 | 86.00 | 58.00 | 86.00 |
| | avg@$k$ | 53.25 | 38.50 | 44.25 | 51.25 | 53.75 | 51.75 | 38.00 | 56.00 |

accuracy (**CR×Acc**). As LongGenBench relies on LLMs for accuracy evaluation, we use Qwen-3-32B (Qwen, 2025a) as the evaluator.

As shown in Table 2, across all evaluated models, FreeKV maintains overall accuracy comparable to or exceeding that of the model with full KV cache. Compared to other methods, FreeKV achieves the best or second-best performance in terms of CR and overall accuracy. For long-generation tasks, RazorAttention with static dropping suffers significant accuracy losses, while RaaS with dynamic dropping demonstrates strong accuracy, likely due to the relative simplicity of the tasks. In addition, we observe repeated output and reduced accuracy for ShadowKV with Qwen-2.5 models, which can be attributed to errors in the reconstructed keys.

**Long reasoning tasks**    Besides the general long-generation tasks where users prompt to generate long-form content, reasoning models like DeepSeek-R1 autonomously generate long thinking processes to solve complex problems. We evaluate reasoning tasks using problems from MATH500, AIME24 and GPQA datasets. For testing, we select 50 problems each from MATH500 and GPQA and use the entire AIME24 dataset. Since the outputs of reasoning models are highly sensitive to random seeds (Hochlehnert et al., 2025), we generate $k = 8$ different samples for each problem. We report two metrics: **pass@**$k$, which measures the likelihood of at least one correct solution among the $k$ samples, and **avg@**$k$, which represents the average accuracy across all $k$ samples.

As shown in Table 3, FreeKV delivers accuracy comparable to models with full KV cache and outperforms other compression methods across most datasets. KV dropping methods such as RazorAttention and RaaS exhibit significant accuracy losses, particularly on AIME24, which involves more complex problems. Moreover, FreeKV consistently outperforms other KV retrieval methods in most cases, demonstrating the effectiveness of its page summaries, softmax-based group consistent selection, and fine-grained correction mechanism.

## 5.3 EFFICIENCY EVALUATION

**Setup**    Our experiments were conducted on an Nvidia A100 40GB GPU, connected with AMD 7302 CPUs via PCIe Gen4. The evaluation covers Qwen-2.5-7B and Llama-3.1-8B models under both long-input (32K input, 512 output) and long-generation scenarios (600 input, 16K output). We set $\tau$ to 0.8 for long-input and 0.9 for long-generation scenarios, with $\mathcal{B} = 2048$ and $\mathcal{S} = \mathcal{W} = 512$. We exclude Quest from the efficiency evaluation because it does not support offloading and fails to complete inference for large batch sizes and long contexts (e.g., batch size 4 with 32K contexts) due to GPU out-of-memory errors.

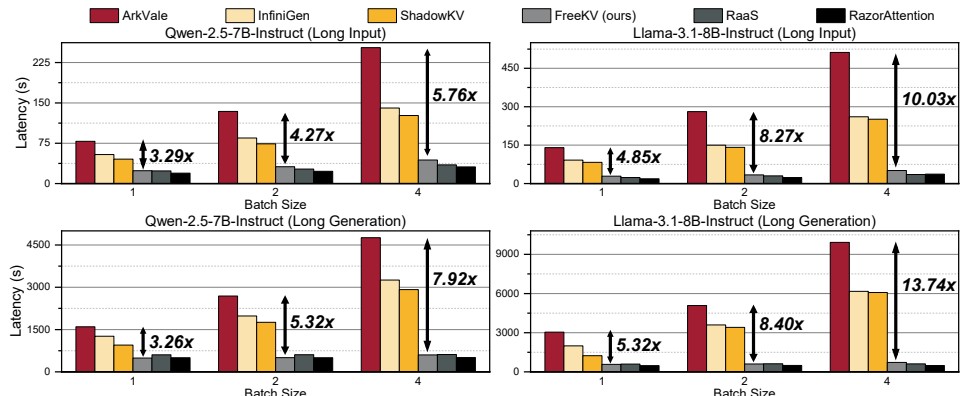

Figure 7: End-to-end latency comparison between KV compression methods.

**End-to-end latency**  As shown in Fig. 7, FreeKV demonstrates significant efficiency gains over SOTA KV retrieval methods, achieving up to 13.7× and 8.4× speedups compared to ArkVale and ShadowKV, respectively. Moreover, FreeKV attains efficiency comparable to dropping methods like RaaS and RazorAttention, which do not involve offloading or recall. The speedups over ArkVale are detailed in Fig. 7. For InfiniGen, FreeKV achieves 3.2× and 5.4× speedups under long-input and long-generation scenarios on Qwen-2.5-7B, and 5.1× and 8.5× on Llama-3.1-8B. The improvements over ShadowKV are comparable to those over InfiniGen, reaching up to 8.4× on Llama-3.1-8B in the long-generation scenario. The improvements become more pronounced for large batch sizes and in long-generation scenarios, where more recall operations are required. In addition, the improvements are amplified for Llama-3.1-8B, which has more KV heads and a larger KV cache compared to Qwen-2.5-7B. Moreover, we present inference latency across different input and output lengths in Appendix C.1, showing that FreeKV consistently achieves substantial speedups under various settings. We also conduct ablation studies on the impact of our efficiency optimizations in Appendix C.2, which demonstrate their effectiveness.

## 6 DISCUSSION

While FreeKV achieves near-lossless accuracy, techniques such as adaptive budgets (Feng et al., 2024) or dynamic budgets with top-$p$ sparsity (Chen et al., 2024b; Lin et al., 2025; Zhou et al., 2025) can be applied orthogonally to further enhance accuracy. In addition, machine learning based methods have been proposed to predict attention patterns for KV cache compression (Yang et al., 2025; Akhauri et al., 2025). However, these methods introduce significant training and runtime overhead and are only effective for long-input scenarios. Moreover, although page-wise selection is found to be less effective for small budgets (Liu et al., 2024; Hooper et al., 2024), learnable block-wise sparsity techniques, applied during pre-training (Yuan et al., 2025; Lu et al., 2025) or post-training (Gao et al., 2025b), show promise in achieving native and optimal page-wise KV cache compression and retrieval.

## 7 CONCLUSION

We present FreeKV, an algorithm-system co-optimization KV retrieval framework that integrates speculative retrieval and fine-grained correction on the algorithm side, as well as hybrid layouts and streamed recall on the system side. FreeKV achieves near-lossless accuracy across various scenarios and models, delivering up to 13× speedup over SOTA KV retrieval methods.

ACKNOWLEDGMENTS

This work is sponsored by the National Natural Science Foundation of China (62472273, 62232015).

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

Table 4: Accuracy results of recall using query vector of the last layer and the last step.

|  | LongBench v2 | MATH500 | AIME24 | GPQA |
|---|---|---|---|---|
| **Query Vector of the Last Layer** | 26.44 | 68.00 | 33.00 | 30.50 |
| **Query Vector of the Last Step** 
 **(Speculative Retrieval w/o Correction)** | 26.63 | 69.00 | 50.00 | 36.00 |

Table 5: Accuracy results of alternatives for achieving group-consistent selection.

|  | LongBench v2 | MATH500 | AIME24 | GPQA |
|---|---|---|---|---|
| **MaxQ** | 26.64 | 70.00 | 51.67 | 38.00 |
| **MeanQ** | 26.64 | 70.00 | 46.67 | 37.00 |
| **MaxQK** | 26.84 | 71.00 | 46.67 | 36.50 |
| **MeanQK** | 26.84 | 70.00 | 51.67 | 34.00 |
| **MaxS** | 26.64 | 69.50 | 47.50 | 36.50 |
| **MeanS** | **26.84** | 70.00 | **52.92** | **39.50** |

## A    DETAILED EXPERIMENTAL SETTINGS

**Baselines**    We consistently set the page size to 32 for FreeKV, Quest, ArkVale, ShadowKV and RaaS. Since the original implementations of RaaS, Quest and InfiniGen are not group-consistent, we adapt them by applying maximum pooling over scores within the group to ensure consistent selection. For ShadowKV, which does not natively support long-generation scenarios, we modify it to update the SVD results every $\mathcal{W}$ generated tokens. Following standard practice, KV cache compression is not applied to the first layer in any of the methods. Other hyperparameters of the baselines, such as the update threshold for RaaS, the SVD rank for ShadowKV, and the skewing rank for InfiniGen, are retained as specified in their original configurations.

The sink size $\mathcal{S}$, local window size $\mathcal{W}$, correction threshold $\tau$ and sampling policies vary depending on the task.

**LongBench v2**    We set $\mathcal{S} = \mathcal{W} = 128$ and $\tau = 0.8$, and apply greedy decoding.

**LongGenBench**    We set $\mathcal{S} = \mathcal{W} = 512$ and $\tau = 0.9$, applying stochastic sampling with a temperature of 0.95, a top-$p$ value of 0.95, and a maximum generation length of 16K following the original setup.

**Reasoning Tasks**    we set $\mathcal{S} = \mathcal{W} = 512$, $\tau = 0.9$ and the maximum generation length to 16K, and apply stochastic sampling with a temperature of 0.6 and a top-$p$ value of 0.95, following the original DeepSeek-R1 setup.

## B    ABLATION STUDIES FOR MODEL ACCURACY

### B.1    RECALL USING THE QUERY VECTOR OF THE LAST LAYER VS. THE LAST STEP

InfiniGen computes skewed query vectors by re-projecting hidden states of layer $l - 1$ to select and recall KV tuples for layer $l$. To demonstrate the superiority of speculative retrieval in FreeKV, which recalls using the query vector of the last decoding step, we adapt InfiniGen to select and recall directly with the last layer's query vector under the same page-wise setting as FreeKV.

Table 4 reports overall accuracy on LongBench v2 and avg@$k$ on reasoning tasks, evaluated with Qwen-2.5-7B-Instruct and DeepSeek-R1-Qwen-7B under the same setting described in Section 5.2. As shown, while using the query vector from the last layer achieves accuracy comparable to speculative retrieval on LongBench v2 and relatively simple reasoning tasks such as MATH500, it incurs substantial accuracy degradation on more challenging reasoning tasks, including AIME24 and GPQA.

Table 6: Accuracy results of alternatives for achieving group-consistent correction.

| | LongBench v2 | MATH500 | AIME24 | GPQA |
|---|---|---|---|---|
| **Max Pooling over Group** $\mathcal{C}_i$ | 26.84 | 70.00 | 53.33 | 39.00 |
| **Mean Pooling over Group** $\mathcal{C}_i$ | 26.84 | 70.00 | 52.92 | 39.50 |

Table 7: Accuracy results of different correction thresholds.

| | LongBench v2 | MATH500 | AIME24 | GPQA |
|---|---|---|---|---|
| $\tau = 0$ **(No Correction)** | 26.63 | 69.00 | 50.00 | 36.00 |
| $\tau = 0.7$ | 26.63 | 70.00 | 50.00 | 36.00 |
| $\tau = 0.8$ | 26.84 | 70.50 | 51.66 | 35.50 |
| $\tau = 0.9$ | 26.84 | 70.00 | 52.92 | 39.50 |
| $\tau = 1$ **(No Speculation)** | 26.84 | 70.00 | 53.00 | 39.50 |

## B.2 ALTERNATIVES FOR ACHIEVING GROUP-CONSISTENT SELECTION

To achieve group-consistent selection, we evaluate max and mean pooling across heads within the same group for (i) query vectors (**Q**), (ii) attention weights between query vectors and page summaries (**QK**), and (iii) normalized attention weights after softmax (**S**).

Table 5 reports overall accuracy on LongBench v2 and avg@$k$ on reasoning tasks, evaluated with Qwen-2.5-7B-Instruct and DeepSeek-R1-Qwen-7B under the same setting described in Section 5.2. We observe that mean pooling over the weights after softmax (**MeanS**) achieves the best overall performance, which is the strategy adopted in FreeKV.

## B.3 EFFECT OF CORRECTION IN SPECULATIVE RETRIEVAL

**Group-consistent correction**  The query vector similarity ($\mathcal{C}_i$) of heads within the same group needs to be unified to enable group-consistent correction. Table 6 reports overall accuracy on Long-Bench v2 and avg@$k$ on reasoning tasks, comparing max and mean pooling over $\mathcal{C}_i$ with Qwen-2.5-7B-Instruct and DeepSeek-R1-Qwen-7B. As shown, both pooling strategies yield similar accuracy, but max pooling triggers more corrections and incurs higher overhead. Therefore, FreeKV adopts mean pooling.

**Correction threshold** $\tau$  Since correction is triggered when $\mathcal{C}_i < \tau$, higher threshold $\tau$ would trigger more correction, theoretically yielding better accuracy with higher costs. Table 7 shows overall accuracy on LongBench v2 and avg@$k$ on reasoning tasks using different $\tau$, with Qwen-2.5-7B-Instruct and DeepSeek-R1-Qwen-7B. As shown, while using $\tau = 0.8$ works well for simple tasks such as LongBench v2 and MATH500, it underperforms on complex reasoning tasks, perhaps due to accumulated errors during reasoning. Therefore, we use $\tau = 0.8$ for long-input scenarios and $\tau = 0.9$ for long-generation scenarios.

## C ABLATION STUDIES FOR INFERENCE EFFICIENCY

### C.1 EFFICIENCY UNDER DIFFERENT INPUT AND OUTPUT LENGTHS

Fig. 8 reports the inference latency of ArkVale and FreeKV across different input and output lengths, using the Llama-3.1-8B-Instruct model under the same settings as Sec. 5.3. In the long-input scenario, FreeKV consistently outperforms ArkVale by 2.7×–5.3×. The relative speedup decreases for longer inputs due to the increasing prefill cost, which is identical for both methods. In the long-generation scenario, FreeKV maintains a stable 5.3× speedup across all output lengths, benefiting from its fixed-size KV budget and efficient recall overlap.

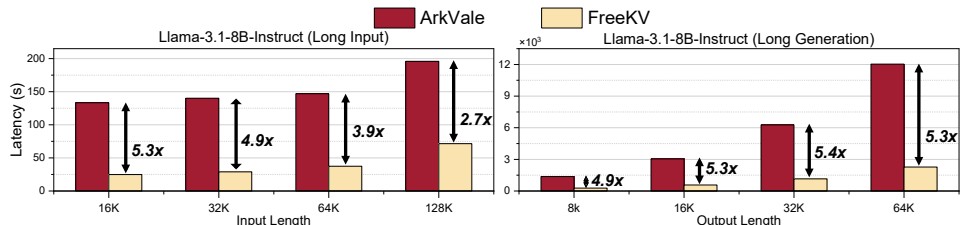

Figure 8: Inference latency for different input and output lengths.

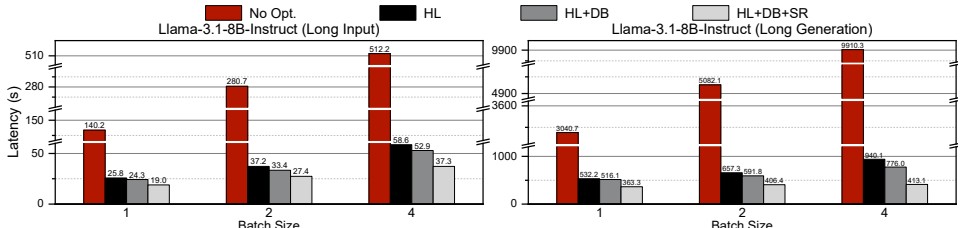

Figure 9: Ablation results for efficiency optimizations.

## C.2 EFFECT OF OPTIMIZATION TECHNIQUES

We present the ablation results of efficiency optimizations applied in FreeKV, including hybrid layouts (HL), double-buffering streamed recall (DB) and speculative retrieval (SR), evaluated using Llama-3.1-8B-Instruct under long-input and long-generation scenarios. As shown in Fig. 9, hybrid layouts, which eliminate fragmented data transfers, contribute the most to the improvements, achieving up to a $10.5\times$ speedup. For a batch size of 4, streamed recall adds a further $1.2\times$ speedup, while overlapping with speculative retrieval provides an additional $1.9\times$ speedup.

While HL alone provides the largest gains, the roles of SR and system-level optimizations are equally critical in our algorithm–system co-optimization framework. The SR algorithm enables pre-selection and prefetching of important KV pages, making it possible to overlap recall operations with ongoing computation. However, as illustrated in the latency breakdown in Fig. 1 (right), recall operations in SOTA KV retrieval methods can be overwhelmingly slow. Therefore, applying speculative retrieval without system-side optimizations results in limited efficiency benefit, as the slow recall operations remain on the critical path and cannot be effectively overlapped. The system-side optimizations (HL and DB) dramatically reduce the latency of recall operations to the same order as other operations, enabling effective overlapping and full latency hiding and allowing practical speedup of speculative recall.

## D GENERALIZATION OF HIGH QUERY SIMILARITY

To validate the generalization of high query similarity, we analyze the query similarity across tasks, model scales, architectures, and training stages, presented in Table 8.

- Tasks: We first measure the query similarity of Qwen-2.5-7B-Instruct across multiple tasks, including LongBench, LongGenBench, AIME24, MATH500, and GPQA. We observe that similarity in the first layer is relatively low, where FreeKV and other compression methods are not applied. Therefore, we report the average similarity across all other layers and decoding steps. As shown in the table, query similarity remains consistently high, around 0.9 for all tasks.

- Model scales: To validate that high query similarity generalizes across model scales, we measure it for Qwen-2.5-1.5B, 3B, 7B, and 14B-Instruct. As shown in the table, query similarity remains consistently high across all scales and tasks, around 0.9, demonstrating that this property is largely model-scale agnostic.

Table 8: Query similarity across various tasks, model scales, architectures, and training stages.

| Model | LongBench v2 | LongGenBench | MATH 500 | AIME24 | GPQA |
|---|---|---|---|---|---|
| Qwen-2.5-7B-Instruct | 0.91 | 0.89 | 0.90 | 0.90 | 0.90 |
| Qwen-2.5-1.5B-Instruct | 0.92 | 0.91 | 0.91 | 0.92 | 0.91 |
| Qwen-2.5-3B-Instruct | 0.89 | 0.89 | 0.90 | 0.90 | 0.90 |
| Qwen-2.5-14B-Instruct | 0.88 | 0.88 | 0.89 | 0.89 | 0.89 |
| Llama-3.1-8B-Instruct | 0.87 | 0.89 | 0.88 | 0.88 | 0.88 |
| Qwen-3-8B | 0.82 | 0.85 | 0.85 | 0.84 | 0.85 |
| Qwen-2.5-7B (Base) | 0.90 | 0.91 | 0.90 | 0.90 | 0.90 |
| DeepSeek-R1-Distill-Qwen-7B | 0.87 | 0.86 | 0.86 | 0.86 | 0.86 |

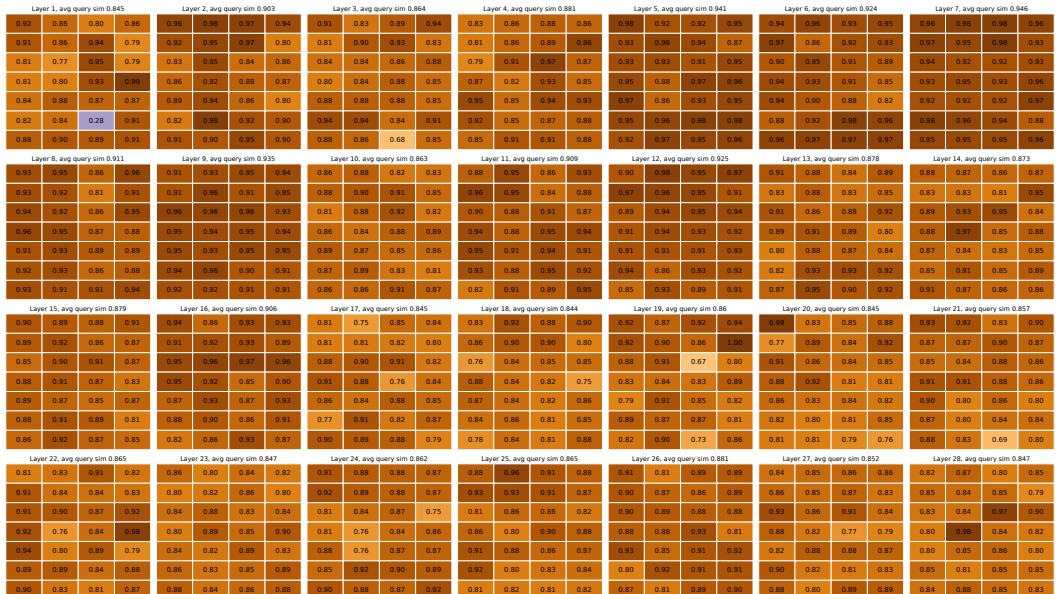

Figure 10: Per-head query similarity of Qwen-2.5-7B-Instruct on LongBench.

- Model architectures: To validate that high query similarity generalizes across model architectures, we measure it for Qwen-2.5-7B-Instruct, Llama-3.1-8B-Instruct, and Qwen-3-8B. As shown in the table, similarity remains consistently high across architectures, ranging from 0.85 to 0.9. The slightly lower similarity in Qwen-3 may be due to its mixed-thinking training recipes.

- Training stages: To validate that high query similarity generalizes across different training stages, we measure it for Qwen-2.5-7B (Base, Pretrain-only), Qwen-2.5-7B-Instruct (SFT & RLHF), and DeepSeek-R1-Distill-Qwen-7B (Long-CoT RL). As shown in the table, similarity remains consistently high across all stages, ranging from 0.86 to 0.9, indicating that this property is robust to variations in training.

In addition, we provide the detailed per-head query similarity of some models and tasks, in Figure 10 and Figure 14. We observe that similarity in the first layer is relatively low, where FreeKV and other compression methods are not applied.

These results demonstrate that high query similarity is a consistent property across tasks, model scales, architectures, and training stages. This robustness supports the general applicability of FreeKV's speculative retrieval mechanism.

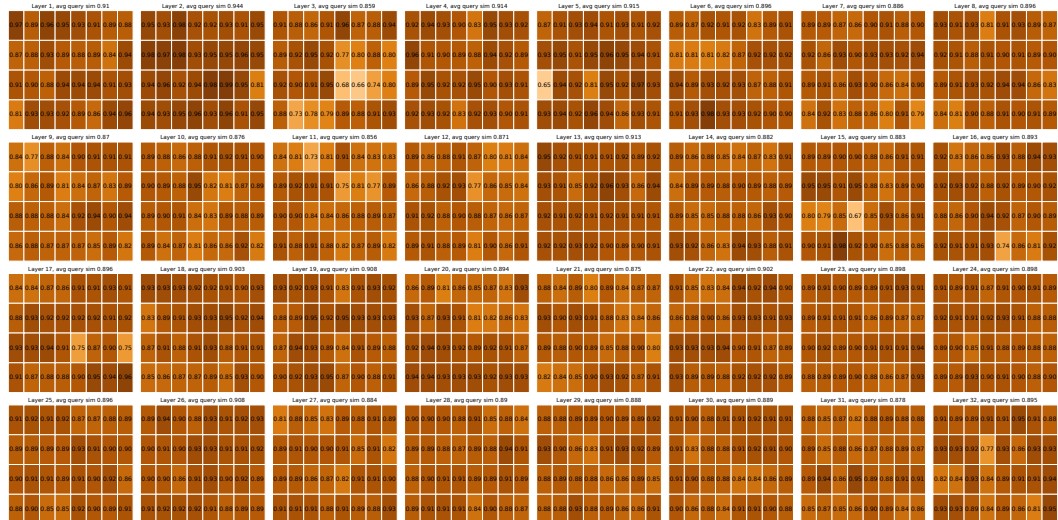

Figure 11: Per-head query similarity of Llama-3.1-8B-Instruct on LongGenBench.

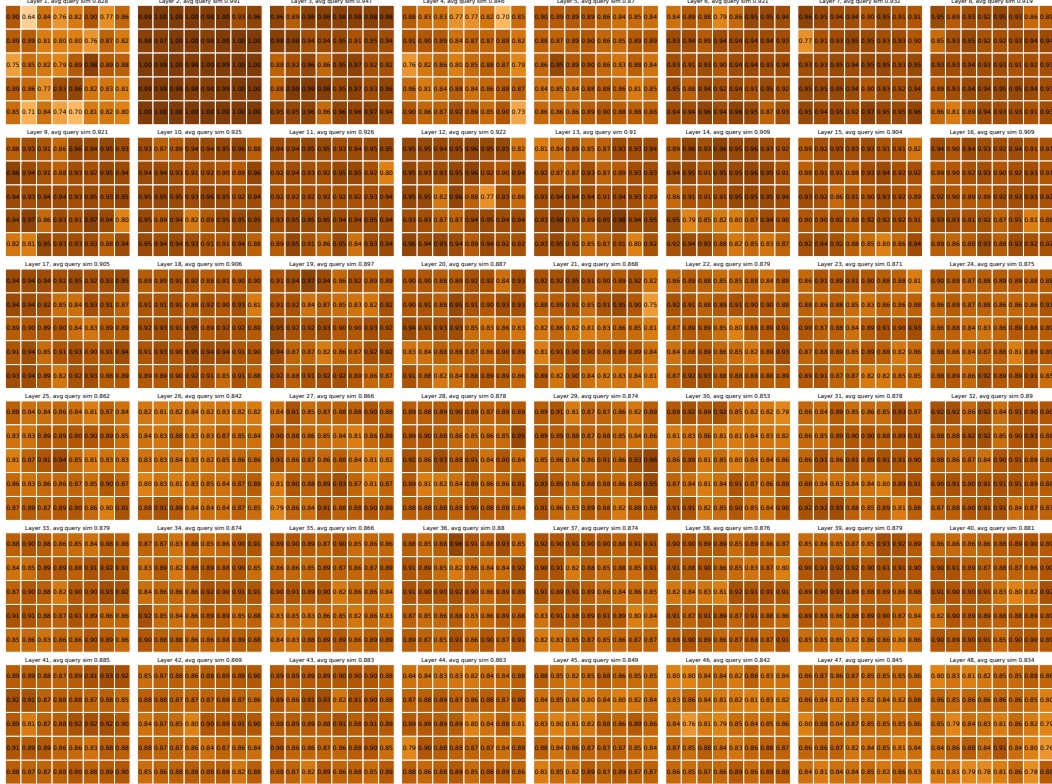

Figure 12: Per-head query similarity of Qwen-2.5-14B-Instruct on MATH500.

# E  CORRECTION RATE ANALYSIS

We report the correction rate, i.e., the fraction of KV heads corrected averaged over decoding steps, across various tasks, models, and thresholds $\tau$.

Following our original setup, we use Llama-3.1-8B-Instruct and Qwen-2.5-7B-Instruct for Long-Bench v2 and LongGenBench, and their DeepSeek-R1 variants for reasoning tasks.

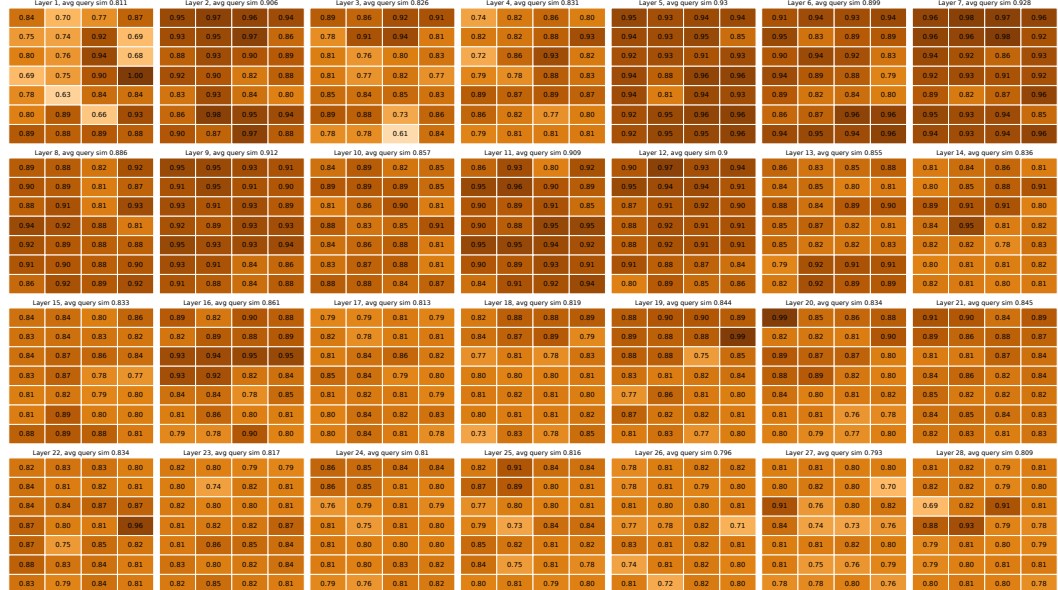

Figure 13: Per-head query similarity of DeepSeek-R1-Qwen-7B on AIME24.

Table 9: Correction rates across different models, tasks and thresholds. Fin stands for that only finished examples are counted.

| | LongBench v2 | LongGenBench | MATH500 | AIME24 | GPQA | AIME24 (Fin.) | GPQA (Fin.) |
|---|---|---|---|---|---|---|---|
| Llama-8B, $\tau = 0.8$ | 0.14 | 0.05 | 0.04 | 0.10 | 0.08 | 0.04 | 0.06 |
| Llama-8B, $\tau = 0.9$ | N/A | 0.17 | 0.20 | 0.43 | 0.35 | 0.19 | 0.29 |
| Qwen-7B, $\tau = 0.8$ | 0.09 | 0.04 | 0.08 | 0.22 | 0.18 | 0.11 | 0.15 |
| Qwen-7B, $\tau = 0.9$ | N/A | 0.22 | 0.17 | 0.52 | 0.47 | 0.29 | 0.41 |

The average correction rates for each task are shown in the Table 9. The rates range from 0.04 to 0.52 and depend on task difficulty. Simpler tasks (LongBench v2, LongGenBench, MATH 500) exhibit lower correction rates, while harder reasoning tasks (AIME24, GPQA) trigger more frequent correction. We also observe that correction is disproportionately higher on unfinished hard examples; when considering only finished cases, correction rates drop substantially.

We also provide per-layer histograms of detailed correction rate distributions across tasks, models, and thresholds, shown in Figure 15 to Figure 19.

# F  LLM USAGE

Large Language Models (LLMs) were only used to aid in the polishing of this work.

It is important to note that the LLM was not involved in the ideation, research methodology, or experimental design. All research concepts, ideas, and analyses were developed and conducted by the authors. The contributions of the LLM were solely focused on improving the linguistic quality of the paper, with no involvement in the scientific content or data analysis.

The authors take full responsibility for the content of the manuscript, including any text generated or polished by the LLM. We have ensured that the LLM-generated text adheres to ethical guidelines and does not contribute to plagiarism or scientific misconduct.

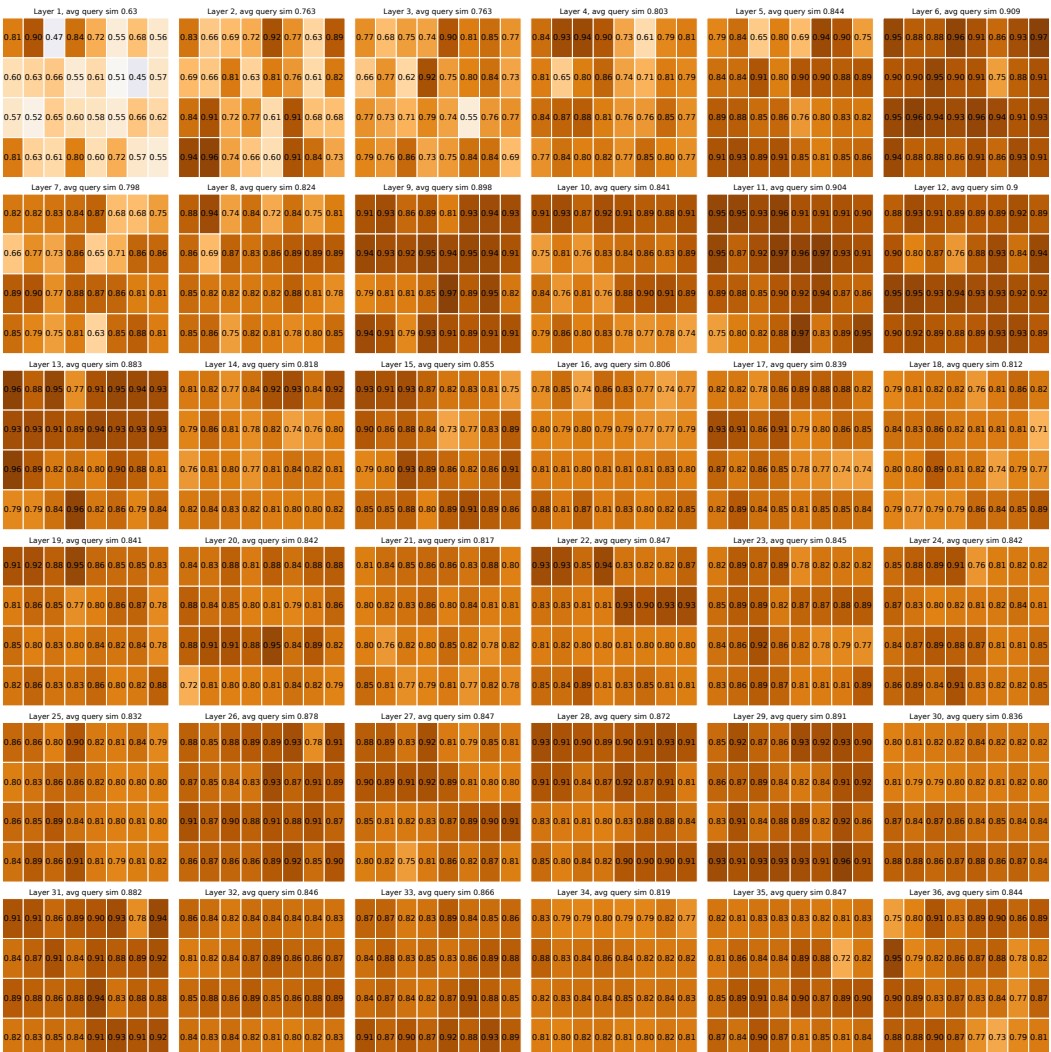

Figure 14: Per-head query similarity of Qwen-3-8B on GPQA.

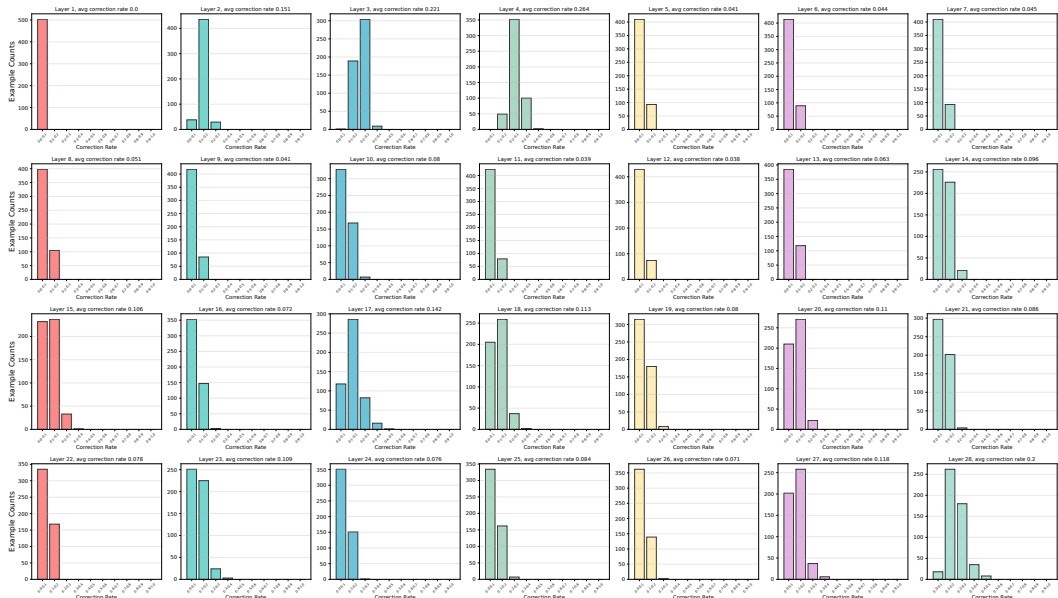

Figure 15: Per-layer distribution of correction rates of Qwen-2.5-7B-Instruct on LongBench with $\tau = 0.8$.

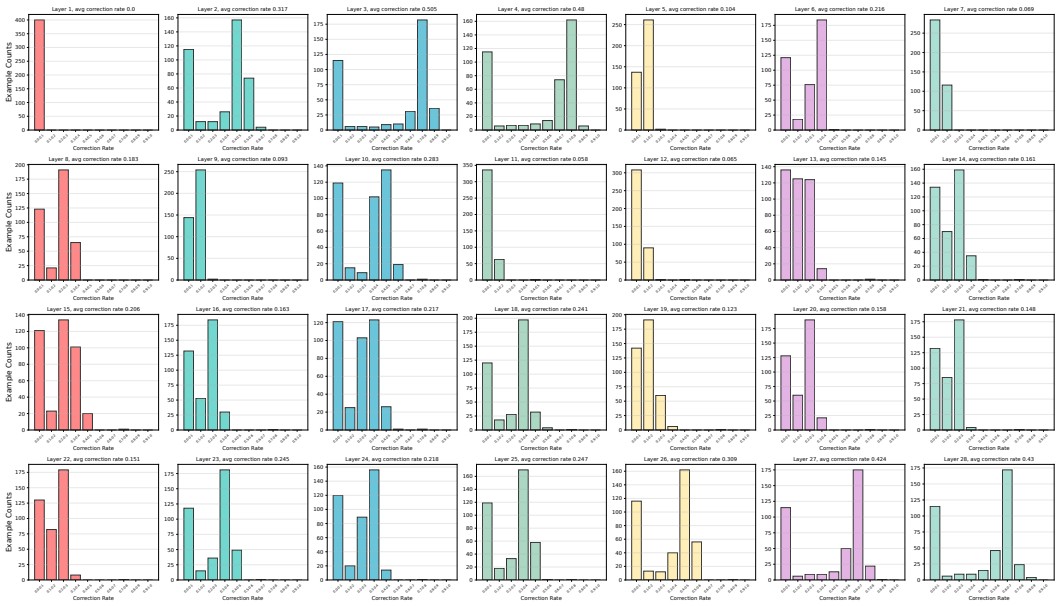

Figure 16: Per-layer distribution of correction rates of Qwen-2.5-7B-Instruct on LongGenBench with $\tau = 0.9$.

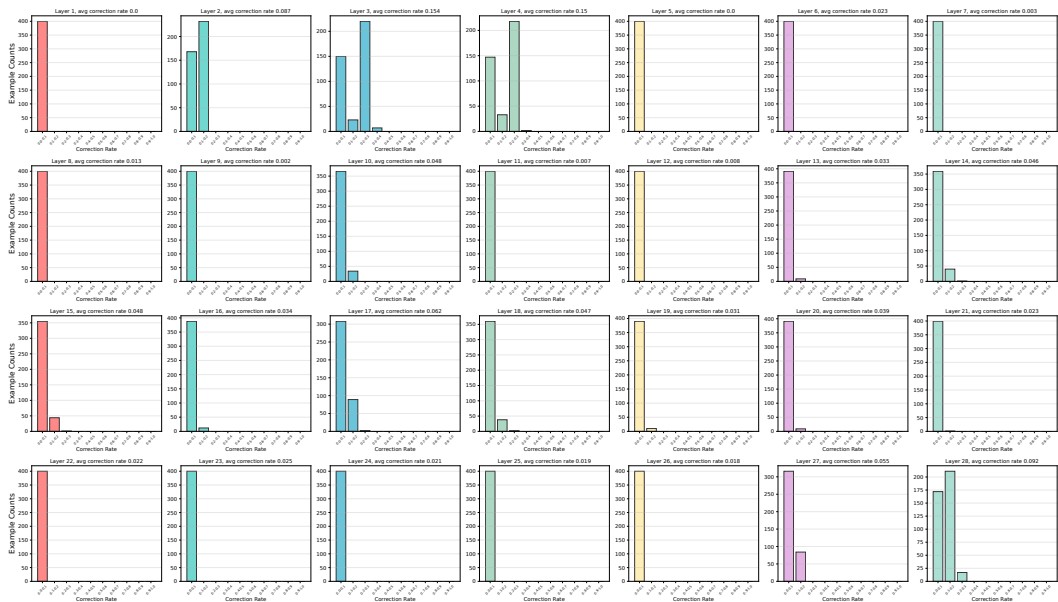

Figure 17: Per-layer distribution of correction rates of Qwen-2.5-7B-Instruct on LongGenBench with $\tau = 0.8$.

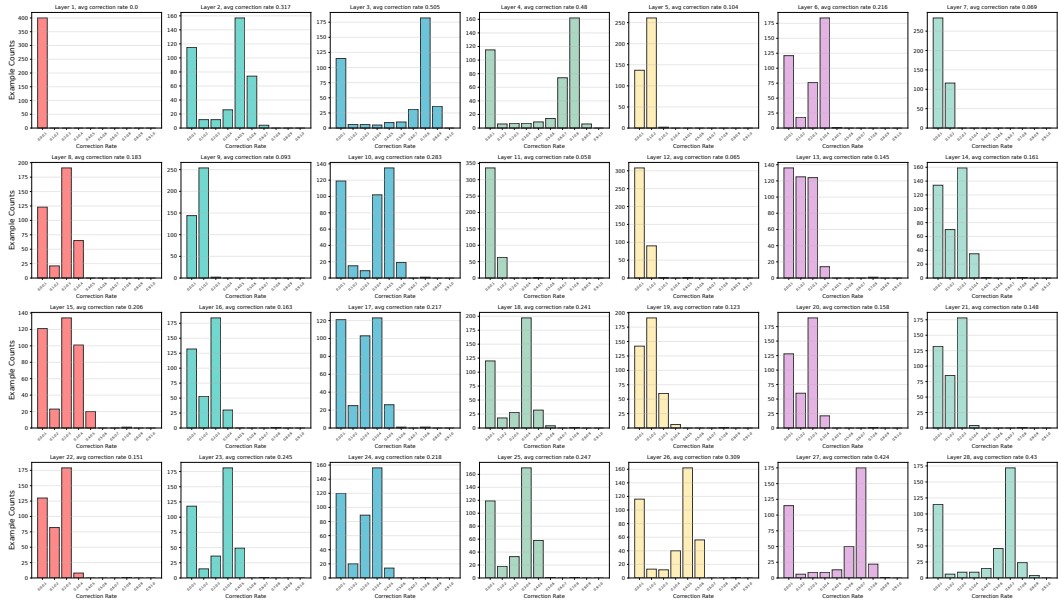

Figure 18: Per-layer distribution of correction rates of Llama-3.1-8B-Instruct on LongGenBench with $\tau = 0.9$.

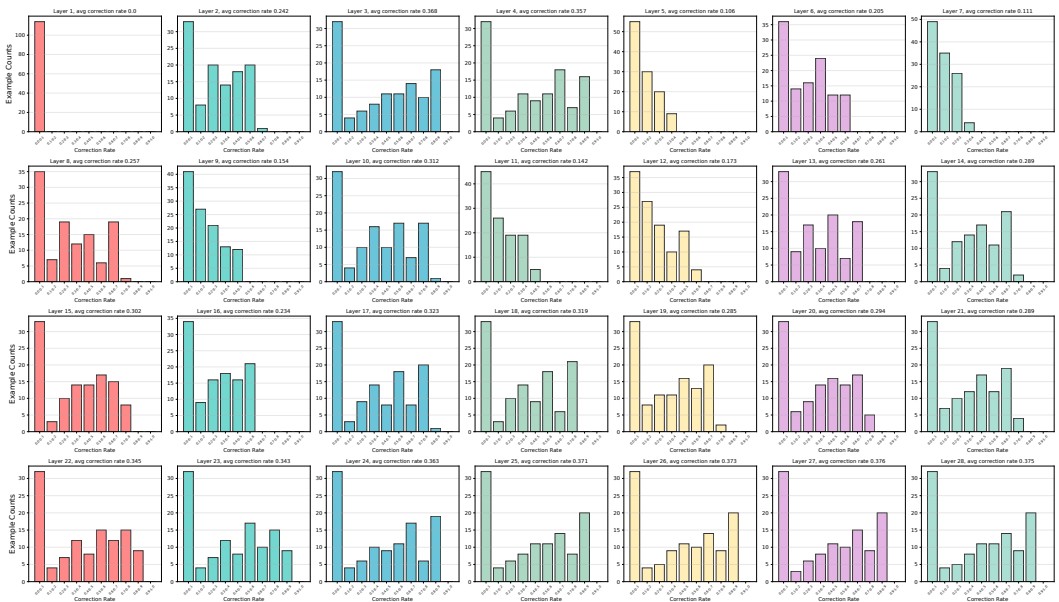

Figure 19: Per-layer distribution of correction rates of DeepSeek-R1-Qwen-7B on three reasoning tasks with $\tau = 0.9$.

