# OpenReview forum: "FreeKV: Boosting KV Cache Retrieval for Efficient LLM Inference"
_ICLR.cc/2026/Conference — ICLR 2026 Poster_

### Official Review · Reviewer_NYAv · 2025-10-20

**Soundness:** 3
**Presentation:** 3
**Contribution:** 3
**Rating:** 8
**Confidence:** 3

**Summary:**

FreeKV introduces a unified algorithm-system co-optimization framework that dramatically accelerates LLM inference (decoding specifically) by eliminating KV cache retrieval latency. It does this through speculative retrieval, which reuses KV caches from the previous decoding step to fully overlap selection and recall with ongoing computation, and head-wise fine-grained correction, which selectively updates only attention heads whose query patterns change. FreeKV achieves up to 13 times faster inference compared with prior KV retrivial methods and better accuracy then KV dropping methods.

**Strengths:**

The paper proposes FreeKV, a speculative KV retrieval framework built on the key insight that query vectors in adjacent decoding steps exhibit high similarity. This observation enables efficient reuse of previous KV selections, and represents a novel and meaningful advancement in accelerating LLM inference (if the observation is empirically validated).

The paper presents the approach with clear organization and technical depth, detailing both the algorithmic design (speculative retrieval and head-wise correction) and the system-level implementation (hybrid layouts and streamed recall). The figures and timelines are well-constructed, making the method intuitive to follow.

The experimental evaluation is comprehensive and convincing, covering multiple models and datasets. The results consistently demonstrate that FreeKV achieves a superior trade-off between efficiency and accuracy compared with existing KV retrieval and KV dropping methods, highlighting both practical impact and generality.

**Weaknesses:**

While the design of FreeKV is primarily motivated by the observed similarity of queries across adjacent decoding steps, the validation of this core assumption remains limited. Section 3.1 provides some preliminary evidence, but the analysis is neither comprehensive across diverse models and prompts nor sufficiently detailed to characterize the factors influencing query similarity. A deeper investigation would strengthen the paper, because of the worry that this method could perform bad on corner case scenarios, which is also interesting to know. For example, examining whether similarity levels correlate with specific tasks, semantic patterns, or model architectures, and how training or fine-tuning methods affect this property. Additionally, it would be valuable to quantify what degree of similarity is necessary to maintain decoding accuracy. Although the appendix includes relevant ablation studies, a concise summary and discussion of these findings in the main text would make the paper’s design justification more convincing.

**Questions:**

What factors could affect the "query similarity" property the paper assumed?

---

> ### Author Response · Authors · 2025-11-20
>
> We sincerely appreciate your constructive feedback on our paper. We incorporate several updates addressing your concerns in our revised version. Below, we provide our detailed responses to each point.
>
> **Q1: Deeper analysis on query similarity**
>
> We agree that a deeper analysis of query similarity strengthens the foundation of FreeKV.
> To validate the **generalization of high query similarity**, we analyze the query similarity across **tasks, model scales, architectures, and training stages**.
>
> - Tasks:
> We first measure the query similarity of Qwen-2.5-7B-Instruct across multiple tasks, including LongBench, LongGenBench, AIME24, MATH500, and GPQA. We observe that similarity in the **first layer is relatively low**, where FreeKV and other compression methods are not applied.
> Therefore, we report the average similarity across all other layers and decoding steps.
> As shown in the table, **query similarity remains consistently high, around 0.9 for all tasks.**
>
> |                      | LongBench v2 | LongGenBench | MATH 500 | AIME24 | GPQA |
> | -------------------- | ------------ | ------------ | -------- | ------ | ---- |
> | Qwen-2.5-7B-Instruct | 0.91         | 0.89         | 0.90     | 0.90   | 0.90 |
>
> - Model scales:
> To validate that high query similarity generalizes across model scales, we measure it for **Qwen-2.5-1.5B, 3B, 7B, and 14B-Instruct**.
> As shown in the table, query similarity remains consistently high across all scales and tasks, around 0.9, demonstrating that this property is largely model-scale agnostic.
>
> |                        | LongBench v2 | LongGenBench | MATH 500 | AIME24 | GPQA |
> | ---------------------- | ------------ | ------------ | -------- | ------ | ---- |
> | Qwen-2.5-1.5B-Instruct | 0.92         | 0.91         | 0.91     | 0.92   | 0.91 |
> | Qwen-2.5-3B-Instruct   | 0.89         | 0.89         | 0.90     | 0.90   | 0.90 |
> | Qwen-2.5-7B-Instruct   | 0.91         | 0.89         | 0.90     | 0.90   | 0.90 |
> | Qwen-2.5-14B-Instruct  | 0.88         | 0.88         | 0.89     | 0.89   | 0.89 |
>
> - Model architectures:
> To validate that high query similarity generalizes across model architectures, we measure it for Qwen-2.5-7B-Instruct, Llama-3.1-8B-Instruct, and Qwen-3-8B.
> As shown in the table, similarity remains consistently high across architectures, ranging from 0.85 to 0.9.
> The slightly lower similarity in Qwen-3 may be due to its mixed-thinking training recipes.
>
> |                       | LongBench v2 | LongGenBench | MATH 500 | AIME24 | GPQA |
> | --------------------- | ------------ | ------------ | -------- | ------ | ---- |
> | Qwen-2.5-7B-Instruct  | 0.91         | 0.89         | 0.90     | 0.90   | 0.90 |
> | Llama-3.1-8B-Instruct | 0.87         | 0.89         | 0.88     | 0.88   | 0.88 |
> | Qwen-3-8B             | 0.82         | 0.85         | 0.85     | 0.84   | 0.85 |
>
> - Training stages:
> To validate that high query similarity generalizes across different training stages, we measure it for Qwen-2.5-7B (Base, Pretrain-only), Qwen-2.5-7B-Instruct (SFT & RLHF), and DeepSeek-R1-Distill-Qwen-7B (Long-CoT RL).
> As shown in the table, similarity remains consistently high across all stages, ranging from 0.86 to 0.9, indicating that this property is robust to variations in training.
>
> |                             | LongBench v2 | LongGenBench | MATH 500 | AIME24 | GPQA |
> | --------------------------- | ------------ | ------------ | -------- | ------ | ---- |
> | Qwen-2.5-7B (Base)          | 0.90         | 0.91         | 0.90     | 0.90   | 0.90 |
> | Qwen-2.5-7B-Instruct        | 0.91         | 0.89         | 0.90     | 0.90   | 0.90 |
> | DeepSeek-R1-Distill-Qwen-7B | 0.87         | 0.86         | 0.86     | 0.86   | 0.86 |
>
> In addition, we provide the detailed per-head query similarity of some models and tasks in the updated Appendix D (Figure 10 to Figure 14) in the paper.
> As shown, the query similarity is consistently high for most of query heads.
>
> These results demonstrate that **high query similarity is a consistent property** across tasks, model scales, architectures, and training stages.
> This robustness supports the general applicability of FreeKV’s speculative retrieval mechanism.

---

> > ### Author Response · Authors · 2025-11-20
> >
> > **Q2: What degree of similarity is necessary to maintain decoding accuracy**
> >
> > As demonstrated in Table 7 in Appendix B.3, using $\tau = 0.8$ for long-input scenarios and $\tau = 0.9$ for long-generation yields near-lossless accuracy. In other words, **speculative retrieval remains accurate when reusing the previous retrieval results for query similarities in the 0.8–0.9 range**. This validates that the similarity levels required for safe speculative reuse are well within the typical similarity observed during decoding.
> >
> > Additionally, in FreeKV, decoding accuracy is preserved regardless of the query similarity, thanks to the **correction mechanism**.
> > If query similarity is very low, speculative retrieval is automatically disabled, and FreeKV falls back to standard KV cache retrieval, ensuring accuracy.
> >
> > Furthermore, given the **consistently high query similarity** observed across tasks, model scales, architectures, and training stages, FreeKV is able to **simultaneously achieve efficiency through speculative retrieval and maintain accuracy via the correction mechanism** in practice.

---

> > > ### Comment · Reviewer_NYAv · 2025-11-23
> > > **Response to the rebuttal**
> > >
> > > The rebuttal addressed my major concerns. I will keep my rating score.

---

> > > > ### Author Response · Authors · 2025-11-24
> > > >
> > > > Thank you for the thoughtful comments and for taking the time to re-evaluate the manuscript. We are glad the rebuttal resolved your major concerns and appreciate your decision to keep the rating score.

---

### Official Review · Reviewer_9HjF · 2025-10-25

**Soundness:** 2
**Presentation:** 2
**Contribution:** 1
**Rating:** 2
**Confidence:** 5

**Summary:**

This paper addresses the efficiency bottleneck of KV cache retrieval in LLM inference and proposes FreeKV, an algorithm–system co-optimization framework. It introduces speculative retrieval to move KV selection and recall off the critical path and fine-grained correction to maintain accuracy. On the system side, it uses hybrid CPU–GPU layouts and streamed recall to reduce data transfer overhead. Experiments show near-lossless accuracy and up to 13× speedup over prior retrieval methods.

**Strengths:**

1. The paper tackles an important and practical problem in LLM serving: efficient KV retrieval under long contexts.
2. FreeKV presents well-motivated system–algorithm co-design with comprehensive experiments covering multiple models and tasks, demonstrating impressive empirical speedups and accuracy preservation.

**Weaknesses:**

1. FreeKV improves runtime efficiency via engineering and overlap techniques. Its algorithmic novelty is incremental over prior work like InfiniGen and ArkVale.
2. The speculative reuse in FreeKV depends on strong query similarity assumptions that may not generalize to all model architectures or reasoning tasks.

**Questions:**

1. quantify how query similarity impacts recall accuracy or establish conditions where correction is triggered.
2. Include ablation studies isolating algorithmic vs. system-side gains, to show which components (speculative reuse, fine-grained correction, hybrid layout) contribute most to the speedup.
3. Strengthen novelty positioning: highlight what design principles or insights make FreeKV different from prior works like InfiniGen and ArkVale, rather than appearing as an engineering integration.

---

> ### Author Response · Authors · 2025-11-20
>
> We sincerely appreciate your constructive feedback on our paper. We incorporate several updates addressing your concerns in our revised version. Below, we provide our detailed responses to each point.
>
> **Q1: Novelty positioning of FreeKV**
>
> The core novelty of FreeKV lies in the **speculative retrieval mechanism**, which exploits **high query similarity across adjacent decoding tokens** to facilitate **efficient pre-selection and prefetching** for KV cache retrieval.
> This approach allows FreeKV to anticipate future KV needs with minimal overhead while maintaining high generalization accuracy.
> By contrast, InfiniGen relies on **offline calibration and additional projections** to leverage inter-layer similarity, introducing extra overhead and limiting adaptability.
>
> Moreover, FreeKV couples this algorithmic insight with **system-level optimizations**, including hybrid KV layouts and streamed recall that enable **full latency overlap** and practical speedups.
> In contrast, ArkVale lacks both speculative retrieval and efficient system integration, causing slow recall on the critical path and poor practical performance.
>
> Together, FreeKV’s design demonstrates a **principled combination of algorithmic insight and system optimization**, rather than a straightforward engineering integration.

---

> ### Author Response · Authors · 2025-11-20
>
> **Q2: Generalization of query similarity**
>
> We agree that a deeper analysis of query similarity strengthens the foundation of FreeKV.
> To validate the **generalization of high query similarity**, we analyze the query similarity across **tasks, model scales, architectures, and training stages**.
>
> - Tasks:
> We first measure the query similarity of Qwen-2.5-7B-Instruct across multiple tasks, including LongBench, LongGenBench, AIME24, MATH500, and GPQA. We observe that similarity in the **first layer is relatively low**, where FreeKV and other compression methods are not applied.
> Therefore, we report the average similarity across all other layers and decoding steps.
> As shown in the table, **query similarity remains consistently high, around 0.9 for all tasks.**
>
> |                      | LongBench v2 | LongGenBench | MATH 500 | AIME24 | GPQA |
> | -------------------- | ------------ | ------------ | -------- | ------ | ---- |
> | Qwen-2.5-7B-Instruct | 0.91         | 0.89         | 0.90     | 0.90   | 0.90 |
>
> - Model scales:
> To validate that high query similarity generalizes across model scales, we measure it for **Qwen-2.5-1.5B, 3B, 7B, and 14B-Instruct**.
> As shown in the table, query similarity remains consistently high across all scales and tasks, around 0.9, demonstrating that this property is largely model-scale agnostic.
>
> |                        | LongBench v2 | LongGenBench | MATH 500 | AIME24 | GPQA |
> | ---------------------- | ------------ | ------------ | -------- | ------ | ---- |
> | Qwen-2.5-1.5B-Instruct | 0.92         | 0.91         | 0.91     | 0.92   | 0.91 |
> | Qwen-2.5-3B-Instruct   | 0.89         | 0.89         | 0.90     | 0.90   | 0.90 |
> | Qwen-2.5-7B-Instruct   | 0.91         | 0.89         | 0.90     | 0.90   | 0.90 |
> | Qwen-2.5-14B-Instruct  | 0.88         | 0.88         | 0.89     | 0.89   | 0.89 |
>
> - Model architectures:
> To validate that high query similarity generalizes across model architectures, we measure it for Qwen-2.5-7B-Instruct, Llama-3.1-8B-Instruct, and Qwen-3-8B.
> As shown in the table, similarity remains consistently high across architectures, ranging from 0.85 to 0.9.
> The slightly lower similarity in Qwen-3 may be due to its mixed-thinking training recipes.
>
> |                       | LongBench v2 | LongGenBench | MATH 500 | AIME24 | GPQA |
> | --------------------- | ------------ | ------------ | -------- | ------ | ---- |
> | Qwen-2.5-7B-Instruct  | 0.91         | 0.89         | 0.90     | 0.90   | 0.90 |
> | Llama-3.1-8B-Instruct | 0.87         | 0.89         | 0.88     | 0.88   | 0.88 |
> | Qwen-3-8B             | 0.82         | 0.85         | 0.85     | 0.84   | 0.85 |
>
> - Training stages:
> To validate that high query similarity generalizes across different training stages, we measure it for Qwen-2.5-7B (Base, Pretrain-only), Qwen-2.5-7B-Instruct (SFT & RLHF), and DeepSeek-R1-Distill-Qwen-7B (Long-CoT RL).
> As shown in the table, similarity remains consistently high across all stages, ranging from 0.86 to 0.9, indicating that this property is robust to variations in training.
>
> |                             | LongBench v2 | LongGenBench | MATH 500 | AIME24 | GPQA |
> | --------------------------- | ------------ | ------------ | -------- | ------ | ---- |
> | Qwen-2.5-7B (Base)          | 0.90         | 0.91         | 0.90     | 0.90   | 0.90 |
> | Qwen-2.5-7B-Instruct        | 0.91         | 0.89         | 0.90     | 0.90   | 0.90 |
> | DeepSeek-R1-Distill-Qwen-7B | 0.87         | 0.86         | 0.86     | 0.86   | 0.86 |
>
> These results demonstrate that **high query similarity is a consistent property** across tasks, model scales, architectures, and training stages.
> This robustness supports the general applicability of FreeKV’s speculative retrieval mechanism.
>
> **Q3: How query similarity impacts recall accuracy and conditions where correction is triggered**
>
> The impact of query similarity on recall accuracy is minimal in FreeKV because **any low-similarity step automatically triggers correction**.
> Specifically, when the similarity between the current query and the previous one falls below the threshold $\tau$, speculative retrieval is disabled and FreeKV performs accurate recall using the current query vector.
>
> The trigger conditions, i.e., the threshold $\tau$ is determined through the sensitivity study in Appendix B.3, which shows that $\tau = 0.8$ is appropriate for long-input scenarios and $\tau = 0.9$ is appropriate for long-generation scenarios.
>  As shown in our updated Appendix E, the resulting correction frequency ranges from **0.04 to 0.52**, which iccurs minor efficiency overhead.

---

> > ### Author Response · Authors · 2025-11-20
> >
> > **Q4: Ablation studies isolating algorithmic vs. system-side gains**
> >
> > We present the ablation results of our efficiency optimizations in Figure 9 (Appendix C.2) in the paper.
> > As shown, hybrid layouts (HL) contribute most to the overall efficiency, and speculative retrieval (SR) provides an **additional 2.3x speedup** beyond the gains from simply applying hybrid layouts.
> > The combination of these optimizations drives FreeKV’s overall performance.
> >
> > Moreover, within FreeKV’s **algorithm-system co-optimization** framework, the roles of speculative retrieval and system optimizations like HL are **equally critical**.
> > The speculative retrieval algorithm enables pre-selection and prefetching of important KV pages, making it possible to overlap recall operations with ongoing computation.
> > However, as illustrated in the latency breakdown in Figure 1 (right), recall operations in SOTA KV retrieval methods can be overwhelmingly slow.
> > Therefore, applying speculative retrieval without system-side optimizations results in limited efficiency benefit, as the slow recall operations remain on the critical path and cannot be effectively overlapped.
> > Our system-side optimizations dramatically reduce the latency of recall operations to the same order of other operations, **enabling effective overlapping and full latency hiding and allowing practical speedup of speculative recall**.

---

> > > ### Comment · Reviewer_9HjF · 2025-11-27
> > >
> > > The authors’ response somehow addresses  my concerns to some extent, but I still have reservations about the generalizability of “query similarity” and the overhead it may introduce. I have raised my score to Weak Reject.

---

> > > > ### Author Response · Authors · 2025-11-28
> > > >
> > > > We thank the reviewer for re-evaluating our work and raising the score.
> > > > We appreciate the opportunity to address your remaining reservations regarding the mechanism's generalizability and the associated overhead.
> > > >
> > > > **1. On the Generalizability of Query Similarity**
> > > >
> > > > Our confidence in the generalizability of query similarity stems not only from our extensive empirical results (Table 8, including 8 models across 3 architectures and varying training stages) but also from the fundamental nature of autoregressive generation.
> > > > In autoregressive decoding, adjacent tokens share the same preceding context. The query vectors, which project this context into the attention space, naturally exhibit smooth transitions rather than abrupt changes. We believe that this is a **general property** of the autoregressive decoder-only Transformer architecture.
> > > >
> > > > Moreover, FreeKV does not require "perfect" similarity to work.
> > > > If the computed similarity falls below the threshold $\tau$, FreeKV disables speculative retrieval for that step and reverts to standard, accurate retrieval.
> > > > This ensures that **occasional outlier low-similarity decoding steps do not degrade accuracy**.
> > > >
> > > > **2. On the Overhead of Query Similarity and Correction**
> > > >
> > > > We clarify that the computation and latency overheads are minimal and are already accounted for in our end-to-end performance results.
> > > > First, calculating cosine similarity for correction detection is a simple $O(d)$ vector operation. Compared to the $O(Ld)$ complexity of the selection and attention, this cost is computationally negligible.
> > > >
> > > > We also collect the correction rate, i.e., the fraction of KV heads corrected averaged over decoding steps, across various tasks, models, and thresholds $\tau$.
> > > > The average correction rates for each task are shown in the table below.
> > > > The rates range from **0.04 to 0.52** and depend on task difficulty.
> > > > Simpler tasks (LongBench v2, LongGenBench, MATH 500) exhibit lower correction rates, while harder reasoning tasks (AIME24, GPQA) trigger more frequent correction.
> > > > We also observe that correction is disproportionately higher on **unfinished** hard examples; when considering only finished cases, correction rates drop substantially.
> > > >
> > > > |                      | LongBench v2 | LongGenBench | MATH 500 | AIME24 | GPQA | AIME24 (Finished Only) | GPQA (Finished Only) |
> > > > | -------------------- | ------------ | ------------ | -------- | ------ | ---- | ---------------------- | -------------------- |
> > > > | Llama-8B, $\tau=0.8$ | 0.14         | 0.05         | 0.04     | 0.10   | 0.08 | 0.04                   | 0.06                 |
> > > > | Llama-8B, $\tau=0.9$ | N/A          | 0.17         | 0.20     | 0.43   | 0.35 | 0.19                   | 0.29                 |
> > > > | Qwen-7B, $\tau=0.8$  | 0.09         | 0.04         | 0.08     | 0.22   | 0.18 | 0.11                   | 0.15                 |
> > > > | Qwen-7B, $\tau=0.9$  | N/A          | 0.22         | 0.17     | 0.52   | 0.47 | 0.29                   | 0.41                 |
> > > >
> > > > We further quantify the correction overhead by measuring **end-to-end inference latency** of FreeKV under controlled correction rates using Qwen-2.5-7B and Llama-3.1-8B.
> > > > The table below reports both absolute latency and normalized slowdowns.
> > > > When correction is applied to all KV heads (rate = 1), latency increases by ~30%.
> > > > Under the typical correction rates observed in our benchmarks (0.04–0.52), the overhead remains modest at **8%–19%**, while substantially improving accuracy compared to the no-correction baseline (see Table 7 in the paper).
> > > >
> > > > | Correction Rate | Long Input, Qwen | Long Input, Llama | Long Generation, Qwen | Long Generation, Llama<br> |
> > > > | --------------- | ---------------- | ----------------- | --------------------- | -------------------------- |
> > > > | 0               | 18.1s **(1x)**    | 20.3s **(1x)**     | 379.5s **(1x)**        | 439.3s **(1x)**             |
> > > > | 0.2             | 19.5s **(1.08x)** | 22.2s **(1.1x)**   | 414.1s **(1.09x)**     | 498.7s **(1.14x)**          |
> > > > | 0.4             | 20.7s **(1.14x)** | 22.5s **(1.11x)**  | 444.8s **(1.17x)**     | 524.6s **(1.19x)**          |
> > > > | 0.6             | 21.9s **(1.21x)** | 23.7s **(1.17x)**  | 466.3s **(1.23x)**     | 543.1s **(1.24x)**          |
> > > > | 0.8             | 22.4s **(1.24x)** | 25.0s **(1.24x)**  | 465.3s **(1.23x)**     | 551.9s **(1.27x)**          |
> > > > | 1               | 22.6s **(1.25x)** | 26.0s **(1.29x)**  | 469.6s **(1.24x)**     | 565.6s **(1.29x)**          |
> > > >
> > > > These results clarify that correction is infrequent for most tasks and introduces only minor overhead even in challenging settings, while providing significant accuracy benefits.

---

### Official Review · Reviewer_rpQv · 2025-10-31

**Soundness:** 3
**Presentation:** 3
**Contribution:** 3
**Rating:** 6
**Confidence:** 3

**Summary:**

The paper proposes FreeKV, an algorithm–system co-optimization framework for efficient KV-cache retrieval in long-context LLM inference. The key idea is to leverage the similarity of query vectors across adjacent decoding steps to enable speculative retrieval — shifting KV selection and recall out of the critical path — combined with fine-grained correction when query deviation occurs. On the system side, FreeKV introduces a hybrid CPU–GPU KV layout (HND on CPU, NHD on GPU) and double-buffered streamed recall to overlap transfers and computation. Experimental results show up to 13× speedup over prior retrieval methods (Arkvale, ShadowKV, InfiniGen) with negligible accuracy loss.

**Strengths:**

1. Clear motivation and strong setup:
The paper provides a well-motivated problem statement supported by preliminary empirical analysis.
2. Novel speculative retrieval mechanism:
The proposed speculative retrieval with fine-grained correction is a conceptually novel idea that effectively breaks the strict dependency between KV selection and query scoring, enabling computation–I/O overlap.
3. Comprehensive and convincing experiments:
The paper evaluates FreeKV across diverse models and tasks (e.g., LongBench, and reasoning datasets) against a wide range of baselines. The results show substantial speedups and near-lossless accuracy, demonstrating both practicality and robustness.

**Weaknesses:**

1. Missing analysis of correction overheads:
While the fine-grained correction mechanism is central to FreeKV’s efficiency–accuracy balance, the paper lacks a quantitative analysis of correction frequency and its impact on latency under different similarity thresholds. Such results would clarify the trade-off between performance and accuracy.
2. Limited study on KV budget sensitivity:
The experiments fix the KV budget B but do not explore how varying B influences accuracy and throughput.

**Questions:**

1. FreeKV’s speculative retrieval design always reuses the previously recalled KV pages for the next token. Intuitively, one might expect cumulative drift or stale-selection errors over long generations, especially in reasoning tasks. However, the reported performance on long-context reasoning excels significantly. Could the authors elaborate on why this drift does not appear to accumulate in practice?
2. Since the KV recall happens right after the (t-1)th token selection, which is far before the KV is used by t, maybe we can even store the KVCache in disk?

---

> ### Author Response · Authors · 2025-11-20
>
> We sincerely appreciate your constructive feedback on our paper. We incorporate several updates addressing your concerns in our revised version. Below, we provide our detailed responses to each point.
>
> **Q1: Correction frequency and impact on latency**
>
> We report the **correction rate**, i.e., the fraction of KV heads corrected averaged over decoding steps, across various tasks, models, and thresholds $\tau$.
>
> Following our original setup, we use Llama-3.1-8B-Instruct and Qwen-2.5-7B-Instruct for LongBench v2 and LongGenBench, and their DeepSeek-R1 variants for reasoning tasks.
>
> The average correction rates for each task are shown in the table below.
> The rates range from **0.04 to 0.52** and depend on task difficulty.
> Simpler tasks (LongBench v2, LongGenBench, MATH 500) exhibit lower correction rates, while harder reasoning tasks (AIME24, GPQA) trigger more frequent correction.
> We also observe that correction is disproportionately higher on **unfinished** hard examples; when considering only finished cases, correction rates drop substantially.
>
> |                      | LongBench v2 | LongGenBench | MATH 500 | AIME24 | GPQA | AIME24 (Finished Only) | GPQA (Finished Only) |
> | -------------------- | ------------ | ------------ | -------- | ------ | ---- | ---------------------- | -------------------- |
> | Llama-8B, $\tau=0.8$ | 0.14         | 0.05         | 0.04     | 0.10   | 0.08 | 0.04                   | 0.06                 |
> | Llama-8B, $\tau=0.9$ | N/A          | 0.17         | 0.20     | 0.43   | 0.35 | 0.19                   | 0.29                 |
> | Qwen-7B, $\tau=0.8$  | 0.09         | 0.04         | 0.08     | 0.22   | 0.18 | 0.11                   | 0.15                 |
> | Qwen-7B, $\tau=0.9$  | N/A          | 0.22         | 0.17     | 0.52   | 0.47 | 0.29                   | 0.41                 |
>
> Beyond averages, we also provide **per-layer histograms** and full correction rate distributions across tasks, models, and thresholds in the updated Appendix E (Figure 15-19).
>
> To quantify overhead, we further measure **end-to-end inference latency** of FreeKV under controlled correction rates using Qwen-2.5-7B and Llama-3.1-8B.
> The table below reports both absolute latency and normalized slowdowns.
> When correction is applied to all KV heads (rate = 1), latency increases by ~30%.
> Under the typical correction rates observed in our benchmarks (0.04–0.52), the overhead remains modest at **8%–19%**, while substantially improving accuracy compared to the no-correction baseline (see Table 7 in the paper).
>
> | Correction Rate | Long Input, Qwen | Long Input, Llama | Long Generation, Qwen | Long Generation, Llama<br> |
> | --------------- | ---------------- | ----------------- | --------------------- | -------------------------- |
> | 0               | 18.1s **(1x)**    | 20.3s **(1x)**     | 379.5s **(1x)**        | 439.3s **(1x)**             |
> | 0.2             | 19.5s **(1.08x)** | 22.2s **(1.1x)**   | 414.1s **(1.09x)**     | 498.7s **(1.14x)**          |
> | 0.4             | 20.7s **(1.14x)** | 22.5s **(1.11x)**  | 444.8s **(1.17x)**     | 524.6s **(1.19x)**          |
> | 0.6             | 21.9s **(1.21x)** | 23.7s **(1.17x)**  | 466.3s **(1.23x)**     | 543.1s **(1.24x)**          |
> | 0.8             | 22.4s **(1.24x)** | 25.0s **(1.24x)**  | 465.3s **(1.23x)**     | 551.9s **(1.27x)**          |
> | 1               | 22.6s **(1.25x)** | 26.0s **(1.29x)**  | 469.6s **(1.24x)**     | 565.6s **(1.29x)**          |
>
> These results clarify that correction is infrequent for most tasks, does not cascade heavily across layers, and introduces only minor overhead even in challenging settings, while providing significant accuracy benefits.

---

> ### Author Response · Authors · 2025-11-20
>
> **Q2: KV budget sensitivity**
>
> We appreciate the reviewer’s suggestion and conduct a KV-budget sensitivity study to evaluate how varying the budget B affects both accuracy and end-to-end efficiency.
>
> The accuracy results for Qwen-2.5-7B-Instruct and DeepSeek-R1-Qwen-7B are summarized in the table below.
> As shown, **budgets of 512 and 1024 lead to substantial accuracy degradation**, particularly on challenging reasoning tasks such as AIME24.
> In contrast, **budgets of 2048 and 4096 deliver accuracy close to Full Attention**.
> However, increasing the budget to 4096 doubles the KV memory footprint and attention computation, and also increases recall cost.
> Based on this trade-off, we select 2048 as the default budget in our experiments, as it offers near–full-attention accuracy while maintaining favorable efficiency.
>
> | Budget         | LongBench v2 | MATH500 | AIME24 | GPQA  | Avg.  |
> | -------------- | ------------ | ------- | ------ | ----- | ----- |
> | Full Attention | 27.44        | 71.75   | 56.66  | 35.75 | 47.90 |
> | 512            | 26.24        | 66.00   | 29.17  | 28.00 | 37.35 |
> | 1024           | 27.04        | 71.00   | 44.16  | 36.50 | 44.67 |
> | 2048           | 26.84        | 70.00   | 52.92  | 39.50 | **47.31** |
> | 4096           | 27.63        | 70.00   | 53.34  | 41.00 | **47.99** |
>
> We further analyze the impact of budget B on end-to-end latency using Qwen-7B.
> The results are shown in the table below.
> As shown, for long-input cases, the impacts of budgets are not significant due to consistent prefill costs and overlapped recall in FreeKV.
> For long-generation scenarios, recall overlap again keeps latency stable across budgets 512–2048, and increasing the budget to 4096 introduces only a 7% latency overhead.
>
> | Budget | Long Input | Long Generation |
> | ------ | ---------- | --------------- |
> | 512    | 23.1s       | 490.4s           |
> | 1024   | 23.1s       | 490.8s           |
> | 2048   | 23.4s       | 495.1s           |
> | 4096   | 24.2s       | 527.2s           |
>
> **Q3: Why the drift does not appear to accumulate?**
>
> We thank the reviewer for raising this insightful point.
> We shared the same concern during the early design of FreeKV that repeatedly reusing previously retrieved KV pages might introduce cumulative drift, especially over long reasoning/generation.
> We experimented with a lightweight calibration strategy that periodically re-prefilled the last N generated tokens with full attention.
> However, our experiments consistently showed that **FreeKV achieves strong long-generation and long-reasoning accuracy without needing any calibration**.
> We attribute the absence of drift accumulation to two key factors:
>
> **(1) Autoregressive decoding naturally attenuates small retrieval errors.**
> As long as the model predicts the next token correctly, the hidden states for the subsequent step realign, preventing retrieval noise from compounding.
> Success of some KV cache compression/reuse methods on reasoning tasks \[1, 2\] can also can also demonstrate such *self-correcting* property .
>
> **(2) FreeKV does not always reuse stale selections.**
> Thanks to the correction mechanism, FreeKV only reuses selections when adjacent steps agree with high confidence.
> When divergence occurs, the correction mechanism recomputes or refines the selection, preventing error propagation.
> Therefore, drift does not originate from speculative retrieval but from rare sparse attention computation.
>
> **Q4: Store the KV cache in disk**
>
> We thank the reviewer for this insightful question.
> Compared to **layer-wise prefetch methods** like InfiniGen, FreeKV’s **token-wise (step-wise) prefetch** provides more room and flexibility for recalling in advance.
> However, if recall is too slow and executed sequentially, it can still **block the inference pipeline**.
> As illustrated in Figure 4(a) in the paper, assuming that if the recall at (Step $i-1$, Layer $l$) can only finish when (Step $i$, Layer $l$) begins, the recall at (Step $i-1$, Layer $l+1$) cannot finish when (Step $i$, Layer $l+1$) begins, and the recall will become the bottleneck again.
>
> A promising direction to alleviate this is to **store KV caches of different layers in separate storage devices**, allowing recalls to execute in parallel across layers.
> Under this setup, slower storage (e.g., disk) could be tolerated without blocking the pipeline.
> We plan to explore this approach in future work.
>
>
> \[1\] Cai et al., R-KV: Redundancy-aware KV Cache Compression for Reasoning Models
>
> \[2\] Chen et al., MemShare: Memory Efficient Inference for Large Reasoning Models through KV Cache Reuse

---

### Official Review · Reviewer_tvZV · 2025-11-02

**Soundness:** 3
**Presentation:** 3
**Contribution:** 3
**Rating:** 6
**Confidence:** 3

**Summary:**

The paper introduces FreeKV, a KV-cache retrieval framework that combines algorithmic ideas—speculative retrieval that reuses pages recalled at the previous step, group-consistent page selection with softmax-pooled scores, and fine-grained head-wise correction triggered by low query-similarity—with a systems design that uses hybrid KV layouts (HND on CPU, NHD on GPU) and double-buffered streamed recall to cut fragmented transfers and overlap CPU to GPU recall with compute. The system keeps page summaries and last-step queries on GPU, stores the full KV pool on CPU in a transfer-friendly layout, and converts layout on-the-fly during recall. Across LongBench v2, LongGenBench, and several long-reasoning tasks, FreeKV matches or nearly matches full-KV accuracy while delivering large end-to-end latency reductions versus prior retrieval methods. Reported speedups reach up to 13.7× vs ArkVale and up to 8.4× vs ShadowKV on A100-PCIe settings, with ablations attributing a major share to the hybrid-layout plus streamed-recall pipeline.

**Strengths:**

* The paper offers a clear algorithm–system co-design in which speculative reuse moves selection and recall off the critical path and head-wise correction restores accuracy only when needed.
* The hybrid HND/NHD layout and double-buffered streamed recall directly target PCIe fragmentation and enable effective overlap, which is a practical systems contribution likely to transfer to real serving stacks.
* The empirical study spans multiple models and benchmarks with detailed settings, and demonstrates near-lossless accuracy relative to full-KV along with large end-to-end speedups, supported by ablations over lengths and components.

**Weaknesses:**

* LongBench v2 spans 8K to 2M tokens, yet the paper truncates inputs to 64K and caps generation to 16K, which leaves the very-long regime underexplored where offloading dominates; please add at least one ≥128K long-input case and one ≥32K long-generation case to validate scaling and stress the recall pipeline.
* Since accuracy relies on an LLM-as-judge (Qwen-3-32B), consider strengthening the evaluation with a larger or multi-judge setup, report inter-judge agreement to calibrate the scores.

**Questions:**

* What is the average correction rate (fraction of KV heads corrected per step) across tasks? How often does correction cascade across layers? A per‑layer histogram and the incremental latency/bytes attributable to correction would clarify the practical cost of robustness.
* In which scenarios does speculative reuse mis‑speculate?

---

> ### Author Response · Authors · 2025-11-20
>
> We sincerely appreciate your constructive feedback on our paper. We incorporate several updates addressing your concerns in our revised version. Below, we provide our detailed responses to each point.
>
> **Q1: Results for ≥128K long-input and ≥32K long-generation cases**
>
> We appreciate the reviewer’s suggestion and agree that evaluating longer input/output lengths is important for understanding the scalability of FreeKV in regimes where offloading dominates.
>
> To this end, we have included extended experiments in Figure 8 (Appendix C.1) with 128K-token inputs and 64K-token generations.
> For Llama-3.1-8B, FreeKV achieves **2.7x–4.9x lower end-to-end latency** than ArkVale across input lengths from 32K to 128K, and improves TPOT (time per output token) by **6x** consistently.
> In the long-generation setting, FreeKV sustains a **stable 5.3x speedup** over ArkVale.
>
> These results demonstrate that FreeKV continues to scale effectively in the ≥128K input and ≥32K generation regimes, validating the robustness of our recall pipeline under long-context workloads.
>
> |                          | 32K Input       | 64K Input       | 128K Input      | 16K Output       | 32K Output        | 64K Output        |
> | ------------------------ | --------------- | --------------- | --------------- | ---------------- | ----------------- | ----------------- |
> | ArkVale, E2E Latency (s) | 140.2           | 147.0           | 195.9           | 3040.7           | 6226.4            | 12792.6           |
> | FreeKV, E2E Latency (s)  | 28.9 **(4.9x)** | 37.5 **(3.9x)** | 71.5 **(2.7x)** | 571.1 **(5.3x)** | 1154.2 **(5.4x)** | 2411.7 **(5.3x)** |
> | ArkVale, TPOT (ms)       | 253.0           | 266.6           | 294.2           | 185.5            | 190.0             | 195.2             |
> | FreeKV, TPOT (ms)        | 41.9 **(6.0x)** | 44.3 **(6.0x)** | 46.1 **(6.4x)** | 34.8 **(5.3x)**  | 35.2 **(5.4x)**   | 36.8 **(5.3x)**   |
>
> **Q2: Larger LLM judge and multi-judge setup**
>
> Among our benchmarks, only LongGenBench relies on an LLM-as-judge.
> We agree that incorporating larger or multiple judge models can further strengthen the reliability of the evaluation.
>
> To address this, we additionally re-evaluated the LongGenBench outputs (generated by Qwen-2.5-7B-Instruct) using state-of-the-art judge LLMs, including **DeepSeek-V3.2-Exp (671B)** and **Kimi-k2-0905-preview (1T)**.
>
> The accuracy scores (CR$\times$Accuracy) from these judges are summarized in the table below.
> Across all judge models, the results remain highly consistent, and **FreeKV consistently achieves near-lossless accuracy compared to Full Attention**.
> This cross-judge agreement supports the robustness of our accuracy findings.
>
> |           | Qwen3-32B | DeepSeek-V3.2-Exp | Kimi-k2-0905-preview |
> | --------- | --------- | ----------------- | -------------------- |
> | Full      | 31.09     | 29.98             | 30.32                |
> | Razor     | 21.48     | 20.28             | 20.79                |
> | RaaS      | 34.40     | 32.04             | 31.13                |
> | Quest     | 25.96     | 24.94             | 25.61                |
> | ArkVale   | 31.79     | 30.62             | 31.12                |
> | ShadowKV  | 11.43     | 10.99             | 11.08                |
> | InfiniGen | 27.67     | 26.17             | 27.05                |
> | FreeKV    | 32.81     | 32.09             | 32.03                |

---

> ### Author Response · Authors · 2025-11-20
>
> **Q3: Correction rate and impact on latency**
>
> We report the **correction rate**, i.e., the fraction of KV heads corrected averaged over decoding steps, across various tasks, models, and thresholds $\tau$.
>
> Following our original setup, we use Llama-3.1-8B-Instruct and Qwen-2.5-7B-Instruct for LongBench v2 and LongGenBench, and their DeepSeek-R1 variants for reasoning tasks.
>
> The average correction rates for each task are shown in the table below.
> The rates range from **0.04 to 0.52** and depend on task difficulty.
> Simpler tasks (LongBench v2, LongGenBench, MATH 500) exhibit lower correction rates, while harder reasoning tasks (AIME24, GPQA) trigger more frequent correction.
> We also observe that correction is disproportionately higher on **unfinished** hard examples; when considering only finished cases, correction rates drop substantially.
>
> |                      | LongBench v2 | LongGenBench | MATH 500 | AIME24 | GPQA | AIME24 (Finished Only) | GPQA (Finished Only) |
> | -------------------- | ------------ | ------------ | -------- | ------ | ---- | ---------------------- | -------------------- |
> | Llama-8B, $\tau=0.8$ | 0.14         | 0.05         | 0.04     | 0.10   | 0.08 | 0.04                   | 0.06                 |
> | Llama-8B, $\tau=0.9$ | N/A          | 0.17         | 0.20     | 0.43   | 0.35 | 0.19                   | 0.29                 |
> | Qwen-7B, $\tau=0.8$  | 0.09         | 0.04         | 0.08     | 0.22   | 0.18 | 0.11                   | 0.15                 |
> | Qwen-7B, $\tau=0.9$  | N/A          | 0.22         | 0.17     | 0.52   | 0.47 | 0.29                   | 0.41                 |
>
> Beyond averages, we also provide **per-layer histograms** and full correction rate distributions across tasks, models, and thresholds in the updated Appendix E (Figure 15-19), which illustrate that correction does not cascade across layers, while on some tasks the correction rates are relatively higher in the starting and ending layers.
>
> To quantify overhead, we further measure **end-to-end inference latency** of FreeKV under controlled correction rates using Qwen-2.5-7B and Llama-3.1-8B.
> The table below reports both absolute latency and normalized slowdowns.
> When correction is applied to all KV heads (rate = 1), latency increases by ~30%.
> Under the typical correction rates observed in our benchmarks (0.04–0.52), the overhead remains modest at **8%–19%**, while substantially improving accuracy compared to the no-correction baseline (see Table 7 in the paper).
>
> | Correction Rate | Long Input, Qwen | Long Input, Llama | Long Generation, Qwen | Long Generation, Llama<br> |
> | --------------- | ---------------- | ----------------- | --------------------- | -------------------------- |
> | 0               | 18.1s **(1x)**    | 20.3s **(1x)**     | 379.5s **(1x)**        | 439.3s **(1x)**             |
> | 0.2             | 19.5s **(1.08x)** | 22.2s **(1.1x)**   | 414.1s **(1.09x)**     | 498.7s **(1.14x)**          |
> | 0.4             | 20.7s **(1.14x)** | 22.5s **(1.11x)**  | 444.8s **(1.17x)**     | 524.6s **(1.19x)**          |
> | 0.6             | 21.9s **(1.21x)** | 23.7s **(1.17x)**  | 466.3s **(1.23x)**     | 543.1s **(1.24x)**          |
> | 0.8             | 22.4s **(1.24x)** | 25.0s **(1.24x)**  | 465.3s **(1.23x)**     | 551.9s **(1.27x)**          |
> | 1               | 22.6s **(1.25x)** | 26.0s **(1.29x)**  | 469.6s **(1.24x)**     | 565.6s **(1.29x)**          |
>
> These results clarify that correction is infrequent for most tasks, does not cascade heavily across layers, and introduces only minor overhead even in challenging settings, while providing significant accuracy benefits.
>
> **Q4: In which scenarios does speculative reuse mis‑speculate?**
>
> In FreeKV, speculative retrieval is triggered when and only when current query similarity exceeds the threshold.
> The key assumption enabling speculative reuse is that queries with high cosine similarity tend to select highly overlapping KV tuples during retrieval.
> Therefore, speculative reuse can mis-speculate in cases where two queries have high cosine similarity but still induce different retrieval results.
> This happens only when $|q_i|$ and $|q_j|$ are differ significantly, due to the following relation.
> $$q_i\cdot k - q_j\cdot k = (q_i-q_j)\cdot k = |q_i - q_j||k|cos(q_i-q_j,k) = |k|cos(q_i-q_j,k)\sqrt{|q_i|^2+|q_j|^2-2|q_i||q_j|cos(q_i, q_j)}$$
> In practice, we observe that query norms are stable across decoding steps, especially in models with q\_norm.
> As a result, such mis-speculation cases are extremely rare.

---

> > ### Comment · Reviewer_tvZV · 2025-11-28
> >
> > Thanks the reviewers for the rebuttal, it has addressed most of my concerns and I decide to keep my score.

---

> > > ### Author Response · Authors · 2025-11-28
> > >
> > > Thank you for your thoughtful feedback and for taking the time to re-evaluate the manuscript. We're pleased that our rebuttal addressed your key concerns.

---

### Author Response · Authors · 2025-11-29
**Summary for Area Chairs and Senior Area Chairs**

**1. Overview of Consensus**
We are pleased that **3 out of 4 reviewers (NYAv, tvZV, rpQv) have given positive ratings (8, 6, 6)**.
The reviewers recognize the importance of efficient KV cache retrieval for long-context inference and praise FreeKV for its:
- **Novel Algorithm Design:** Successfully proposing speculative retrieval based on query similarity and introducing fine-grained correction to handle outliers efficiently.
- **Effective Algorithm-System Co-design:** Integrating algorithmic insights with solid system optimizations to achieve substantial practical speedups.
- **Strong Empirical Results:** Achieving up to **13× speedup** with near-lossless accuracy across diverse benchmarks and models.


**2. Key Rebuttal Updates**

During the rebuttal, we successfully addressed the constructive feedback from the reviewers with extensive new experiments:

- **Generalization of Query Similarity (Addressing 9HjF, NYAv):** We provided the overall and detailed per-head query similarity analysis across **8 models, 4 tasks, and 3 architectures**  in the updated Appendix D. The results demonstrate that high query similarity is a consistent intrinsic property across tasks, model scales, architectures, and training stages. This robustness supports the general applicability of FreeKV’s speculative retrieval mechanism.
- **Correction Rates and Overhead (Addressing tvZV, rpQv):** We quantified the correction rates (ranging from **0.04 to 0.52**) and demonstrated that the correction overhead is minor (**8%–19%**) relative to the substantial gains in accuracy.
- **Scaling to Extreme Lengths (Addressing tvZV):** We extended experiments to **128K-token inputs** and **64K-token generations**. FreeKV consistently achieves **2.7x–4.9x lower latency** and a **6x improvement in TPOT** compared to ArkVale, proving scalability in extreme regimes.

**3. Response to the Negative Reviewer (9HjF)**

Reviewer 9HjF initially raised concerns about novelty, query-similarity generalization, and missing ablations.
We strengthened the novelty positioning, added results demonstrating the generality of query similarity, and included detailed ablation studies.
Seven days after our reply, the reviewer increased the score from 2 to 4, though still expressing reservations about the query similarity and the overhead.
To address these remaining concerns, we provide more details on the generalizability of query similarity and further clarify that FreeKV does not rely on perfect similarity. We also include results showing that both correction and selection overheads are minor.

---

### Meta-Review · Area_Chair_PSam · 2025-12-31

**Summary:**

This paper proposes FreeKV, an algorithm–system co-optimization framework for improving KV cache retrieval efficiency in long-context LLM inference. The work introduces speculative retrieval based on query similarity, combined with fine-grained correction, and integrates these ideas with system-level optimizations such as hybrid CPU–GPU KV layouts and streamed recall. Reviewers generally agree that the problem addressed is important and timely, and that the paper demonstrates strong empirical performance with substantial speedups and near-lossless accuracy. While one reviewer remained skeptical about the novelty and generalizability of the approach, the majority of reviewers found the contributions meaningful and the empirical evaluation convincing after the rebuttal. Overall, the paper was viewed as a solid systems-oriented contribution with practical impact.

**Reviewer Concerns:**

The reviewers' primary concerns focused on the following aspects:

(1) The novelty and general applicability of the proposed speculative retrieval mechanism compared to existing key-value (KV) retrieval approaches, particularly its dependence on assumptions about query similarity;

(2) Insufficient validation of query similarity across diverse models, tasks, and operating regimes, as well as an inadequate characterization of correction frequency and associated computational overhead;

(3) The method’s scalability to extremely long contexts and the robustness of the evaluation protocol—especially the reliance on LLM-as-a-judge metrics.

In their rebuttal, the authors addressed these concerns through a suite of substantial additional experiments. These included extended evaluations at input lengths of 128K and generation lengths of 64K, comprehensive analyses of query similarity across multiple models and architectures, detailed statistics on correction rates, and cross-validation using alternative judging strategies. While these enhancements satisfied most reviewers, one remained skeptical regarding the algorithmic novelty of the approach and its long-term generalizability.

**Reviewer Scores:**

The reviewer scores reflect an overall positive but not unanimous assessment. Reviewer NYAv rated the paper highly (8) and indicated that the rebuttal addressed their major concerns. Reviewers tvZV and rpQv provided moderately positive evaluations (both 6), acknowledging the strength of the algorithm–system co-design and empirical results, and maintained their scores after the rebuttal. Reviewer 9HjF was significantly more critical, initially rating the paper low and later increasing the score to a weak reject (4) after the rebuttal, while continuing to express reservations about novelty and generalizability. Overall, the majority of reviewers are supportive, with one dissenting reviewer, and the scores converge toward acceptance.

---

### Decision · Program_Chairs · 2026-01-26

Accept (Poster)